



# Learning about precipitation orographic enhancement from snow-course data improves water-balance modeling

Francesco Avanzi[1], Giulia Ercolani[1], Simone Gabellani[1], Edoardo Cremonese[2], Paolo Pogliotti[2], Gianluca Filippa[2], Umberto Morra di Cella[2,1], Sara Ratto[3], Hervè Stevenin[3], Marco Cauduro[4], and Stefano Juglair[4]

[1]CIMA Research Foundation, Via Armando Magliotto 2, 17100 Savona, Italy
[2]Climate Change Unit, Environmental Protection Agency of Aosta Valley, Loc. La Maladière, 48-11020 Saint-Christophe, Italy
[3]Regione Autonoma Valle d'Aosta, Centro funzionale regionale, Via Promis 2/a, 11100 Aosta, Italy
[4]Direzione Operativa Operations, C.V.A. S.p.A., Via Stazione 31, 11024 Châtillon, Italy

**Correspondence:** Francesco Avanzi (francesco.avanzi@cimafoundation.org)

**Abstract.**

Precipitation orographic enhancement depends on both synoptic circulation and topography. Since high-elevation headwaters are often sparsely instrumented, the magnitude and distribution of this enhancement remain poorly understood. Filling this knowledge gap would allow a significant step ahead for hydrologic-forecasting procedures and water management in general.
Here, we hypothesized that spatially distributed, manual measurements of snow depth (courses) could provide new insights into this process. We leveraged 11,000+ snow-course data upstream two reservoirs in the Western European Alps (Aosta Valley, Italy) to estimate precipitation orographic enhancement in the form of lapse rates and consequently improve predictions of a snow-hydrologic modeling chain (Flood-PROOFS). We found that Snow Water Equivalent (SWE) above 3000 m ASL was between 2 and 8.5 times higher than recorded cumulative seasonal precipitation below 1000 m ASL, with gradients up to 1000
mm w.e. km$^{-1}$. Enhancement factors estimated by blending precipitation-gauge and snow-course data were quite consistent between the two hydropower headwaters (median values above 3000 m ASL between 4.1 and 4.8). Including blended gauge-course lapse rates in an iterative precipitation-spatialization procedure allowed Flood-PROOFS to remedy underestimations of both SWE above 3000 m ASL (up to 50%) and importantly precipitation vs. observed streamflow. Runoff coefficients based on blended lapse rates were also more consistent from year to year that those based on precipitation gauges alone (standard
deviation of 0.06 and 0.19, respectively). Thus, snow courses bear a characteristic signature of orographic precipitation, which opens a window of opportunity for leveraging these data sets to improve our understanding of the mountain water budget. This is all the more important due to their essential role in supporting water security and ecosystem services worldwide.



# 1 Introduction

Orographic precipitation is a critical driver of the Earth's water budget (Jiang, 2003), particularly as it affects amount and distribution of snowpack at high elevations and thus freshwater supply and water security during the warm season (Serreze et al., 1999; Bales et al., 2006; Viviroli et al., 2007b; Blanchet et al., 2009; Mott et al., 2014; Sarmadi et al., 2019). On shorter time scales, orographic precipitation may contribute generating floods (Buzzi et al., 1998; Galewsky and Sobel, 2005; Panziera et al., 2015), as well as trigger landslides and avalanches (Roe, 2005). A feature of both stratiform and convective systems

(Roe, 2005), orographic precipitation introduces sharp transitions between wet, windward and dry, leeward slopes that are ubiquitous across continents (rain shadows, see Houston and Hartley, 2003; Anders et al., 2006; Galewsky, 2009; Viale and Nuñez, 2011; Da Ronco et al., 2020). These mechanisms shape local climates, ecosystems, and societies (see Michalet et al., 2003; Roe, 2005; Poschlod, 2015, and references therein). Thus, better understanding how precipitation interacts with elevation has important implications within and beyond Geosciences.

Orographic precipitation has already been subject of extensive research (Bonacina, 1945; Sarker, 1966; Alpert, 1986; Barros and Kuligowski, 1997; Smith and Barstad, 2004; Roe, 2005; Smith, 2006; Rotunno and Houze, 2007; Allamano et al., 2009; Avanzi et al., 2015; Napoli et al., 2019; Ruelland, 2020). The emerging consensus is that distribution, intensity, and duration of orographic precipitation depend on three modulating factors (Rotunno and Houze, 2007): synoptic-circulation patterns, intensity of mesoscale lifting along slopes, and timing of water condensation through convection and turbulence. Interactions

across these three primary factors are notoriously elusive, with cloud microphysics, local terrain heterogeneity, and boundary-layer thermodynamics challenging the somewhat naive notion that precipitation increases with elevation (see Roe, 2005, for a review). For example, Napoli et al. (2019) have showed a saturation-like effect in orographic enhancement above ∼1000 m ASL in the Alps, associated with a monotonous increase in annual-precipitation variability with elevation. This saturation effect was also highlighted by Blanchet et al. (2009) for mean and maximum snowfall across Switzerland. Avanzi et al. (2015)

observed increased variability in extreme-precipitation quantiles with elevation and a reverse orographic effect for annual maximum precipitation of short duration.

This complex nature of precipitation orographic enhancement means that some of its fundamental aspects are still poorly understood, especially across headwaters with complex terrain. This includes magnitude and seasonality of precipitation gradients, how they vary across the landscape during specific events, or how they interact with temperature and relative humidity

to determine precipitation phase (Harpold et al., 2017; Avanzi et al., 2020a). These knowledge gaps are exacerbated by precipitation measurements in snow-dominated headwaters being prone to large errors, such as wind-driven undercatch and snow plugging (Rasmussen et al., 2012; Avanzi et al., 2014). These issues have progressively discouraged the deployment of measurement stations in snow-dominated regions unless frequent maintenance and ground truthing is performed – which is very rare due to its high costs and logistical constraints. Meanwhile, radar or satellite-based measurements in mountain terrain are

challenged by complex topography, radar-beam shielding, and ground echoing (Germann et al., 2006). Despite substantial efforts in recent years to achieve more reliable precipitation measurements (such as the WMO SPICE initiative, see Nitu et al., 2018), the World's water towers (Viviroli et al., 2007b) remain largely ungauged.





Better understanding precipitation distribution across mountain landscapes is not only an open question for fundamental research, but also for operational forecasting (Frei and Isotta, 2019). Precipitation spatialization in mountain-hydrology models

is generally performed through assigning *a-priori* precipitation-lapse rates (e.g., see Bergström., 1992; Viviroli et al., 2007a; Markstrom et al., 2015) or interpolating precipitation-gauge data with various degrees of geostatistical complexity (Frei and Schär, 1998; Daly et al., 2008; Isotta et al., 2014; Foehn et al., 2018; Frei and Isotta, 2019). While cross-validation accuracy of these spatialization methods is high at measurement sites (e.g., mean absolute errors for monthly precipitation below ∼12 mm in Daly et al., 2008), gauge-based spatialization approaches cannot fully overcome the lack of signal at high elevations.

For instance, the seminal work by Frei and Schär (1998) considered an advanced distance-weighting scheme for spatializing daily precipitation across the European Alps, but the vast majority of measurement sites was located below ∼2500 m ASL. In the US, the Parameter-elevation Relationships on Independent Slopes Model (PRISM, see Daly et al., 2008) adopts a weighted climate–elevation relationship based on physiographic similarity, but Zhang et al. (2017) showed that this approach underestimates both precipitation totals above ∼2400 m ASL and the seasonal precipitation lapse rate in the central Sierra

Nevada, California. It follows that predicting precipitation above this "precipitation-gauge line" at ∼2500 m ASL will always imply some degrees of extrapolation.

A largely unexplored solution to grasp precipitation gradients above the precipitation-gauge line is provided by snow-course data. These are spatially distributed measurements of snow depth (HS) and optionally snow density ($\rho_S$) performed over transects of various extent (generally 1+ km) to estimate snow-water resources across the landscape and inform water-supply

forecasting or other applications (see e.g. Hart and Gehrke, 1990; Rice and Bales, 2010). Snow-course data are routinely collected in various regions of the world, such as the western US (Pagano et al., 2004), Norway (Skaugen et al., 2012), or Finland (Lundberg and Koivusalo, 2003), and have frequently been used to develop and evaluate snow-hydrologic models (Jost et al., 2009), snow-mapping algorithms (Dressler et al., 2006; Margulis et al., 2016), or satellite-retrieval methods (Metsämäki et al., 2005).

In the present study, we hypothesized that snow courses bear an additional, characteristic signature of seasonal orographic enhancement, which can be leveraged to fill the gap in precipitation measurements above the precipitation-gauge line and improve hydrologic-model predictive skills as a result. In order to verify this hypothesis, we focused on two high-elevation, hydropower catchments in the western European Alps (Aosta Valley, NW Italy) where 10,000+ snow-course data points have been collected for water-supply forecasting since 2008. We combined these data with ground-based precipitation to first inves-

tigate the relationship between seasonal precipitation totals below the precipitation-gauge line and peak-accumulation snow-course water equivalent (SWE) above 3000 m ASL, and thus derive a climatology of lapse rates obtained by merging these two data sets (blended lapse rates). We then leveraged these blended lapse rates to develop an iterative, two-step spatialization procedure of precipitation that accounts for seasonal orographic effects in addition to daily precipitation variability below the precipitation-gauge line. Third, we evaluated this spatialization procedure using streamflow measurements as well as opera-

tional snow-hydrologic modeling (S3M and Continuum, see Silvestro et al., 2013; Laiolo et al., 2014).



## 2 Data

### 2.1 Study area

Aosta Valley (one of the twenty Italian administrative regions) is located at the north-western edge of the Italian peninsula (Figure 1, a). Embraced by some of the highest peaks in the Alps (Mont Blanc, 4808 m ASL; Monte Rosa, 4634 m ASL;

and Gran Paradiso, 4061 m ASL), Aosta Valley is a typical inner-Alpine valley with marked rain shadows (Isotta et al., 2014). Annual precipitation totals can be as high as ∼1600-1800 mm across the south-eastern, windward slopes or the Mont-Blanc area in the north-western corner of the region, and lower than 600 mm in the central valley. Such comparatively low precipitation totals, coupled with pronounced temperature gradients, make this region prone to droughts and the associated vegetation stress (Cremonese et al., 2017). Precipitation is year-round, with somewhat bimodal seasonality and prevalent peaks

in spring and fall (Crespi et al., 2018). Its topographic imbalance in precipitation distribution and the associated marked orographic gradients make Aosta Valley an ideal region for the present study. About 134 km$^2$ out of 3261 km$^2$ of Aosta Valley are covered by glaciers (4%), meaning this is the most glacierized region in Italy (Smiraglia et al., 2015; Patro et al., 2018).

We focus on two hydropower catchments for testing our research hypothesis: Valpelline (VP) and Beauregard in Valgrisenche (VG, see Figure 1, a and c). The total drainage area of Beauregard is ∼110 km$^2$, 14% of which are covered by glaciers (11.4

km$^2$); elevation ranges from ∼1800 m ASL at the outlet to ∼3500 m ASL. Valpelline spans both a larger extent and a higher elevation range than Beauregard; total drainage area is ∼130 km$^2$, while minimum and maximum elevation are 1539 and 3934 m ASL, respectively. About 12.4 km$^2$ of the drainage area of Valpelline is glacierized (9.5%). Beauregard and Valpelline are located on two opposite sides of Aosta Valley and thus have different prevalent aspect (northwards and westwards, respectively). Both systems are composed by a reservoir and a number of auxiliary intakes, which we lumped together as a single catchment.

Both catchments are ungauged with regard to precipitation or snow-depth automatic weather stations (Figure 1, a and c).

### 2.2 Data

We employed (1) weather and snow data from the network of monitoring stations of the regional authority (https://cf.regione. vda.it/portale_dati.php, visited on June 6, 2020), (2) reconstructed streamflow data for the two hydropower catchments of Beauregard and Valpelline, and (3) high-elevation, manual peak-snow-depth snow courses. The study period covered water

years 2008 through 2019 based on data availability, with a specific focus on water years 2017 through 2019 when most of the snow-course data were collected. We define a "water year" as the period between September 1 and the following August 31, using the calendar year in which it ends (e.g., water year 2019 went from September 1 2018 to August 31 2019). We used September to August rather than October to September (which is frequent elsewhere) because snow accumulation in this region may start as early as September.

Weather data comprised hourly air temperature, relative humidity, incoming-shortwave-solar-radiation, wind, and total-precipitation data at up to ∼100 (temperature), ∼50 (relative humidity), ∼30 (incoming shortwave solar radiation), ∼60 (wind), and ∼80 (total precipitation) measurement points. The actual number of available measurement points changed from day to day because of potential malfunctioning or communication issues. Data were collected, processed, quality-checked, and stored



by the civil-protection regional authority (data access and geometry of the monitoring network: http://presidi2.regione.vda.it/
str_dataview, accessed on June 12, 2020). Quality checks were based on a mixture of automatic filtering and visual screening.

Figure 1, a reports the location of precipitation measurement points used in the present study, while Figure 1, c highlights
that these precipitation gauges cover only low-to-medium elevations, with the highest one at ∼2700 m ASL. Considered
precipitation gauges comprise both heated and unheated sensors, but the hourly precipitation-spatialization technique used
in the following automatically excludes unheated sensors when they are above the rain-snow transition line (Section 3.2).
Regarding undercatch, previous work by the author team has found that the best-suited correction in this region is that by
Allerup et al. (1997). Applying this correction corresponds to gaining 5% to 15% of total precipitation compared to using
non-corrected precipitation data.

The dataset of the regional authority also comprises hourly snow-depth data at ∼50 locations (see Figure 1, a). Measurement
technique is based on ultrasonic ranging, with precision of a few cm (Ryan et al., 2008; Avanzi et al., 2014). While the average
elevation of these snow-depth stations is higher than that of the precipitation gauges, locations above ∼2700 m ASL remain
largely ungauged (Figure 1, c). Snow cover is also routinely monitored by a cooperative consortium collecting manual samples
of snow depth and density across the whole region, with ∼weekly to monthly revisit times for the same measurement plot.
These manual, periodical data were not directly employed in the present study given their discontinuous nature, but we did
use interpolated SWE maps based on combining these manual, periodical data and the ∼50 automatic snow-depth stations
(see details in Section 3). These maps were produced by the local Environmental Protection Agency (ARPA VdA) and were
used as an assimilation source for our snow-hydrologic forecasting chain (Section 3.2). Figure S1 shows an example of all
input samples used by ARPA VdA for the period 1-8 March 2020. Note that these manual, periodical measurements are not
classifiable as snow courses, because they consist of stand-alone measurements in open sites.

Reconstructed streamflow data for the closure sections of Valpelline and Beauregard were provided by Compagnia Val-
dostana delle Acque (CVA), the company managing both hydropower systems. These estimates were based on measurements
of inflow to the plants and changes in reservoir storage, with a proprietary reconstruction method. The uncertainty of this dataset
has never been fully quantified, but such a reconstruction approach corresponds to the standard method used for unimpaired-
flow estimation in other regions of the world (Avanzi et al., 2020b). Annual-runoff totals for the closure sections of Valpelline
and Beauregard according to these reconstructed data (∼1000-1500 mm) are consistent with other gauged sections in the
region. Data were daily and ranged from water year 2008 to 2019, consistent with weather and snow-course data.

Snow-course data were available for five areas of interest, including the two hydropower catchments of Valpelline and
Beauregard and three much smaller hydropower catchments (Cignaga, Gabiet, Goillet, see Figure 1, a). Because of logistical
constraints, only a subset of these five areas of interest was sampled every year, with a markedly larger dataset starting from
water year 2017 (see Table 1 for an inventory). Data were collected around peak-accumulation day, which changed from year to
year and from catchment to catchment as assessed by ARPA VdA based on weather forecasts and snow-accumulation patterns
(see again Table 1 for survey dates). Snow-depth measurements were taken every 50 to 100 m along transects of several
kilometers, which spanned the whole elevation gradient from the local snow line to the catchment drainage divide (see Figure
1, b for an example of these transects for Beauregard and Figures S2 to S12 for a detail of all transects).





Mean elevation of these snow courses was much higher than the elevation captured by the precipitation-gauge and snow-depth-sensor networks (often above 3000 m ASL). The location of transects across each catchment was chosen to capture a variety of physiographic characteristics, while the number of transects for each catchment-year depended on available resources. Snow depth was measured using manual probes, and location of each measurement was recorded using a portable GPS with a precision on the order of meters. Snow-course data for each area of interest were accompanied by a few measurements of bulk-snow density at representative locations, which were averaged to provide a reference estimate for each survey and hence derive SWE.

## 3  Methods

### 3.1  Estimating blended gauge-course lapse rates

We derived blended precipitation-gauge-snow-course lapse rates for Beauregard (water years 2017 through 2019) and Valpelline (water years 2008 through 2013, 2015 through 2019) by first detecting the onset of the snow season for each catchment and each water year as the first hour with at least 20 cm of snow on the ground for a mid-elevation, nearby snow-depth sensor (red squares in Figure 1, elevation was ∼1860 and 1970 m ASL for the snow depth sensor of Beauregard and Valpelline, respectively). We then accumulated hourly precipitation between this onset date and the snow-course date for every precipitation gauge in the same valley of each hydropower catchment – this was done separately for each water year. We finally derived orographic-precipitation enhancement factors for each valley and water year by dividing seasonally cumulative precipitation at gauges and average SWE above 3000 m ASL by seasonally cumulative precipitation at the lowest-elevation precipitation gauge in the same valley; these adimensional enhancement factors measure the magnitude of orographic-precipitation elevation gradients, regardless of seasonal-precipitation totals.

Blended precipitation-gauge-snow-course lapse rates were computed as a least-square-error regression fit between elevation and these enhancement factors. Although snow-course data were also available for other three study areas (Figure 1 and Table 1), these were too small compared to the respective valleys for deriving robust precipitation lapse rates. We therefore calibrated blended precipitation-gauge-snow-course lapse rates only using data from Beauregard and Valpelline (separately, see Section 3.3 for details on the use of the additional courses in this paper).

The general assumption behind blended precipitation-gauge-snow-course lapse rates is that snow-course measurements above 3000 m ASL are representative of total precipitation fallen at those elevations from the onset of the snow season through the snow-course date. In other words, such blended lapse rates assume that the snowpack above 3000 m ASL behaves as a natural precipitation gauge, with no significant mass loss throughout the accumulation season due to snowpack runoff, evaporation, or sublimation. In this framework, accumulating seasonal precipitation since the first hour with at least 20 cm of snow on the ground aimed at capturing precipitation totals for the bulk of the accumulation season, while excluding early-season snowfall events that might result in complete or partial depletion of the snowpack.

While no continuous-time measurement of SWE was available to validate the 3000m-elevation threshold above which to compute average snow-course SWE, and while prescribing a constant threshold for all water years necessarily neglects inter-



annual variability in weather, Hantel et al. (2012) have found snow-line elevations across the Alps on the order of ∼800 m ASL in winter and ∼3000 m ASL in summer (period 1961-2010). Thus our chosen threshold can be used to assume absence of significant snowmelt before at least May.

Because we computed blended lapse rates using seasonally cumulative precipitation and peak SWE data, these lapse rates are representative of *winter*-precipitation gradients. Correctly capturing these seasonal gradients is vital for estimating peak snow-cover distribution and amount and thus forecast summer water supply (e.g., see Pagano et al., 2004; Harrison and Bales, 2016), although precipitation gradients for specific storms may significantly diverge from the observed seasonal lapse rates. Likewise, these lapse rates are not necessarily representative of summer-storm elevation gradients; in this region as well as
across the Alps in general, summer storms are mostly convection-driven (Giorgi et al., 2016), a process that is rare during winter and therefore cannot be fully captured by peak-season SWE measurements.

### 3.2  Spatialization of precipitation based on blended lapse rates

Blended precipitation-gauge-snow-course lapse rates developed in Section 3.1 were used to design an iterative, two-step precipitation-spatialization procedure accounting for orographic effects above the precipitation-gauge line. The ultimate goal
of designing such a spatialization procedure was twofold: on the one hand, we aimed to confirm whether annual-precipitation totals obtained by blended precipitation-gauge-snow-course lapse rates agreed with annual reconstructed runoff, especially in terms of annual runoff coefficients. On the other hand, we aimed to assess whether blended precipitation lapse rates could improve hydrologic predictions (Section 3.3).

To this end, we employed the operational snow-hydrologic forecasting chain Flood-PROOFS, as validated in this region by
Laiolo et al. (2014). Flood-PROOFS consists in automatic spatialization-downscaling procedures for weather input data, a distributed, pixel-based snow model (S3M), a distributed, pixel-based hydrologic model (Continuum), and snow-depth mapping algorithms used for snow-data assimilation (Rebora et al., 2006; Boni et al., 2010; Laiolo et al., 2014; Silvestro et al., 2013). In this paper, we forced Flood-PROOFS with historical data, which corresponds a standard hydrologic simulation in reanalysis mode. The implementation of Flood-PROOFS considered in this paper runs with a spatial resolution of 120 m; the computa-
tional domain covers the entire Aosta-Valley region. More details about Flood-PROOFS's parametrizations and spatialization techniques can be found in the Supporting Information, Section S1.

Similar to other snow-hydrologic models, precipitation spatialization in Flood-PROOFS relies on in-situ precipitation measurements. In the current spatialization procedure, these precipitation measurements are interpolated using a modified-Kriging approach called GRISO (Random Generator of Spatial Interpolation from uncertain Observations, see Pignone et al., 2010;
Puca et al., 2014). The most significant asset of GRISO is that interpolated precipitation for pixels including a precipitation gauge will maintain the same value as that measured by that precipitation gauge (in other words, measurements at precipitation-gauge locations are preserved during interpolation). The covariance structure for each precipitation gauge is dynamical, while precipitation-field values far from all precipitation gauges tend either towards the mean of the precipitation field observed by gauges, or towards zero. In this paper, we chose the second option, but also set an influence radius for each precipitation gauge





equal to 20 km following previous validations of GRISO in Aosta Valley. No pixel of the study area was thus "far enough" from all gauges for this choice to be relevant.

This one-step precipitation-spatialization procedure assumes that measurements taken by the precipitation-gauge network are representative of the overall range of variability in precipitation across the study domain. As we outlined in the Introduction and will further show in Section 4, this assumption does not necessarily hold true in mountain regions that straddle

the precipitation-gauge line, because interpolated precipitation-fields will likely underestimate precipitation totals due to orographic effects missed by the ground-based measurement network. We overcame this issue by developing a modified, two-step GRISO approach as follows. For each time step of interest (in our case, each hour), GRISO was first run using precipitation gauges alone (GRISO1). Second, interpolated-precipitation values at select pixels above 2700 m ASL were enhanced according to their elevation and the seasonal winter enhancement-factor profile calibrated in Section 3.1. Third, GRISO was re-run using

as input the measurements from the physical precipitation gauges and estimates at these select pixels (GRISO2). In this two-step procedure, these orographically enhanced precipitation estimates act as virtual precipitation gauges at high elevations (see location in Figure S13), with orographic enhancement being informed by snow-course measurements at peak accumulation.

High-elevation pixels were selected by first defining a regular grid with spacing equal to 5% of the longitudinal and latitudinal range of the study area, and then taking as candidate locations for these virtual gauges the nodes of this grid. Second, we filtered

out any candidate virtual gauge with elevation below 2700 m ASL as well as those falling outside the study area. Figure S13 shows that the final location of these virtual precipitation gauges is coherent with the orography of out study region, and complements the spatial coverage of the physical precipitation-gauge network (Figure 1).

### 3.3    Evaluating blended precipitation-gauge-snow-course lapse rates from a water-balance perspective

We evaluated precipitation estimates informed by blended precipitation-gauge-snow-course lapse rates by comparing predic-

tions of Flood-PROOFS using GRISO1 vs. those using GRISO2 (see Section 3.2). The evaluation period was water years 2017 to 2019, since these three water years saw a peak in evaluation-data availability (particularly snow-course data). Although water years 2017 to 2019 were also used to calibrate the blended precipitation-gauge-snow-course lapse rates and therefore this was not a fully independent evaluation, doing so was necessary given the lack of snow-course data before 2017 for one of the hydropower catchments (Beauregard) and the need for considering as many years as possible for lapse-rate calibration to

capture interannual variability.

Input-data maps were first used to force the snow model of Flood-PROOFS, S3M, and generate hourly equivalent-precipitation fields that were then used as an input for the hydrologic model Continuum; equivalent precipitation is the pixel-wise sum of rainfall, snowpack runoff, and glacier runoff – if any (see Section S1 for details on these models). S3M can be run in two different modes, that is, only relying on weather inputs (Open-Loop run), or assimilating SWE information from independent

sources (Full-Assim run). SWE is assimilated as both weekly maps produced by ARPA VdA through interpolation of manual measurements according to physiographic features (see Section 2.2) and daily maps produced within Flood-PROOFS by training a multilinear regression across concurrent ultrasonic-snow-depth-sensor measurements (predictand) and physiographic features like elevation, slope, and aspect (predictors). In the present study, evaluation of peak-SWE predictions by S3M against



snow courses was carried out with reference to both Open-Loop and Full-Assim simulations, to disentangle the impact of pre-
cipitation spatialization from that of data assimilation. Simulations of the complete Flood-PROOFS chain (S3M + Continuum)
were only performed in Full-Assim mode because of computational-time constraints.

As we will show in Section 4, assimilated snow maps – and in particular those derived from snow-depth sensors – suffer
from a similar bias as that of precipitation at elevations above the snow-depth-sensor line (see Figure 1). This bias is likely
due to the snow-depth-sensor network being skewed toward representing mid-elevation snowpack. Assimilating these biased
snow maps would largely nullify the potentially positive effect of enhancing orographic effects in GRISO2, so we developed a
correction factor by first recalibrating the snow-depth multilinear-regression model with snow-course data in addition to snow-
depth-sensor data. This recalibration was performed for each week when snow-course data were available between water years
2017 and 2019, and considering all five areas of interest were snow-course data were collected to maximize variety of data. The
mean value of the ratio between recalibrated snow-depth maps and the original ones was then used as a multiplicative factor for
original maps to remedy for high-elevation biases. This correction was only estimated for snow-depth-sensor-based maps both
because preliminary assessments showed that they are the major source of bias compared to weekly SWE maps, and because
they are assimilated daily and as such play a much more important role than weekly SWE maps in driving Flood-PROOFS'
accuracy.

We focused on three evaluation exercises: first, we compared basinwide estimated precipitation according to GRISO1 and
GRISO2 with measured reconstructed streamflow, hence runoff coefficients; second, we ground-truthed peak-SWE predictions
by Flood-PROOFS's snow model (S3M) forced using GRISO1 vs. GRISO2 against snow-course data; third, we compared
cumulative daily streamflow predicted by Flood-PROOFS's hydrologic model (Continuum) forced using GRISO1 vs. GRISO2
against reconstructed streamflow. The first and third evaluation exercises had a traditional water-balance perspective; they
determined whether estimated precipitation with and without orographic enhancement can explain annual total runoff and its
seasonal patterns (both important targets for water-supply forecasting). The second exercise assessed the impact of orographic-
precipitation enhancement on the simulation of snow storage, an intermediate prediction target between precipitation and runoff
with significant implications beyond hydrology (e.g., avalanche forecasting, glacier mass balance).

## 4 Results

### 4.1 Water-balance climatology

Average monthly precipitation across gauges in the valleys of Beauregard and Valpelline (variable $\hat{P}$, in mm) was bimodal,
with peaks in November and May (Figure 2, a, reference period was water year 2009 through 2019 due to earlier gaps in $\hat{P}$
for Beauregard, Figure S14, a). Monthly precipitation was similar between the two valleys (mean difference -2.5 ±7.1 mm).
Nonetheless, precipitation in Beauregard was up to 15 mm higher than in Valpelline during fall and winter (September to
March) and up to ∼6 mm lower during summer (April to August), which highlighted that Beauregard may be more exposed to
winter storms coming from the Mediterranean Sea, with Valpelline being more affected by summer storms from the Atlantic
Ocean. This tallies with the bimodal and the summer-dominated precipitation regimes of the southern and northern sides of



the Alps, respectively (Frei and Schär, 1998; Isotta et al., 2014). However, we stress that $\hat{P}$ only accounts for precipitation gauges, meaning precipitation totals above the precipitation-gauge line remained unaccounted. Annual $\hat{P}$ in both valleys was consistent from year to year: $\sim 700 \pm 84$ mm and $\sim 730 \pm 94$ mm in Valpelline and Beauregard, respectively (Figure 3, a and S14, a).


According to the reference snow-depth sensors used in Section 3.1 to accumulate winter precipitation, the average snow season started in October in both Beauregard and Valpelline (Figure 2, b, reference period is water years 2008 through 2017 due to later gaps in snow-depth data for Valpelline, see Figure 3, b). The end-of-season date generally occurred in May, with both catchments being exposed to late snowfall events even in June. In contrast with precipitation, peak snow depth showed a

remarkable interannual variability (standard deviation of maximum annual snow depth was 31 and 77 cm at Beauregard and Valpelline, respectively), with three of the four water years with shallow snowpacks occurring between 2014 and 2019 (Figure 3, b and S14, b). During some of these shallow-snowpack water years, the snow cover was ephemeral at these snow-depth sensor sites (e.g., water years 2017 at Beauregard, Figure S15, b). Monthly snow depth at the reference sensor of Valpelline was significantly higher than that of Beauregard (Figure 2, b), which we explain because the former is at a higher elevation of

the latter ($\sim 100$ m).

Reconstructed streamflow in both catchments was highly seasonal, with minimum flow during winter and maximum flow in June, when precipitation, snow melt, and ice melt overlap (Figure 2, c, reference period is water years 2008 through 2019). Monthly streamflow was similar between the two catchments (mean difference was -0.11 $\pm$ 18 mm), with the only exception that streamflow in Beauregard was higher (up to $\sim$20 mm) between November and June and lower (up to $\sim$45 mm) between

July and October than in Valpelline. This is likely connected to average monthly precipitation in Beauregard being higher and lower than in Valpelline during winter and summer, respectively (see Figure 2, a). A second argument is favor of Beauregard and Valpelline being hydrologically similar catchments was the similarity in precipitation-runoff relationship (Figure 3, d), that is, the fundamental rule relating annual precipitation to annual runoff (Saft et al., 2016; Avanzi et al., 2020b). Compared to precipitation climatology, reconstructed streamflow in both catchments showed comparatively large interannual variability

(Figure 3, c and S14, c), owing to both precipitation and climate variability.

In summary, ground-based precipitation, streamflow, and snow-depth sensor data showed that water-supply generation in both these two hydropower catchments is fundamentally cryosphere-dominated. They also showed that ground-based sensor data located across low- and mid-elevations are largely insufficient to grasp the full water balance of these cryosphere-dominated headwaters, as also demonstrated by (1) annual $\hat{P}$ being systematically smaller than the corresponding annual

streamflow totals (Figure 3, d), and (2) peak monthly snow depth occurring in February in both catchments. While no continuous-time measurement of snow-depth and SWE was available above $\sim$2700 m ASL (Figure 1), general consensus in the Alps as well as our experience is that peak-SWE date occurs around April 1 or later in high-elevation, Alpine catchments (Marty et al., 2017).





## 4.2 Precipitation vs. SWE orographic gradients

We report in Figure 4, a and b, examples of precipitation-gauge vs. snow-course orographic gradients obtained in 2018 for Beauregard and Valpelline, respectively. At Beauregard, precipitation gauges recorded a positive, but mild precipitation lapse rate – on the order of $\sim$250 mm km$^{-1}$. At Valpelline, the lapse rate recorded by precipitation gauges was even smaller, $\sim$75 mm km$^{-1}$. The orographic trend recorded by gauges agreed with GRISO1 (see again Figure 4, a and b), which was expected given that GRISO1 used precipitation gauges as a starting point to distribute precipitation across the landscape. Snow-course

data drew a substantially different picture from precipitation gauges, with SWE sharply increasing with elevation (Figure 4, a and b): $\sim$1000 and 567 mm w.e. km$^{-1}$ in Beauregard and Valpelline, respectively. Thus, peak SWE close to (or above) 3000 m ASL in 2018 was 2-3 times total winter precipitation measured by precipitation gauges below the precipitation-gauge line.

Examining all water years for which snow-course surveys were available confirmed that these surveys yield much larger orographic gradients than precipitation gauges in both hydropower catchments (Figure 5, a and b and Figures S15 through S25,

where missing panels imply that some of the information needed to perform this comparison was missing for that water year). In particular, precipitation gradients based on gauges hardly exceeded 200 mm w.e. km$^{-1}$, whereas snow-course gradients were often higher than 400 mm w.e. km$^{-1}$ and reached values as high as 1000 mm w.e. km$^{-1}$; snow-course-based gradients were particularly high at Beauregard compared to Valpelline. This sharp increase in snow accumulation with elevation was consistent across water years and was generally underestimated by both snow maps assimilated by Flood-PROOFS in proximity of the

snow-course surveys (Figure 5, a and b and S15 to S25); note that these independent maps do take into account physiography. This is another piece of evidence that ground-based sensor data located across low- and mid-elevations do not capture the complete range of variability in snow distribution at high elevations.

Snow-course-based orographic gradients increased with average snow depth above 3000 m ASL (correlation coefficient $\rho = 0.67$, see Figure 5, c), whereas the correlation between these gradients and snow-course survey date was much weaker ($\rho = $

$0.2$, see Figure 5, d). Moreover, the correlation of snow-course orographic gradients with average snow depth above 3000 m ASL was statistically significant, while that with snow-course survey date was not (p-value of 0.02 and 0.52, respectively). Thus, the choice of survey date had a limited impact on the quantification of snow-course orographic gradients, which suggests that these gradients may preserve themselves through time.

## 4.3 Orographic-enhancement factors

Snow-course measurements of peak SWE above 3000 m ASL were between $\sim$2 and $\sim$8.5 times and between $\sim$4 and $\sim$5.5 times winter cumulative precipitation at the lowest-precipitation gauge of Valpelline and Beauregard valleys, respectively (Figure 6). Interannual variability was significant and partially driven by snowpack amount, with the four highest enhancement factors being associated with water years with low or medium snowpack (see again Figure 6). However, this was not systematic, since the five lowest enhancement factors were recorded during years with mixed characteristics (two with low snowpack, one

with medium snowpack, and two with high snowpack). This tallies with Figure 5, c, where only some dependency between snowpack amount and orographic gradients was found.





Contrary to snow courses, precipitation-gauge-based enhancement factors at low- and mid-elevations showed little to none orographic trends and less interannual variability. Also, they never exceeded 2, meaning precipitation at precipitation-gauge locations was (at best) twice that at the lowest measurement point in the same valley (Figure 6). At Valpelline, a substantial amount of these precipitation-gauge-based enhancement factors were even lower than 1, meaning seasonal cumulative precipitation at intermediate elevations was often lower than that at the lowest-elevation gauge in the valley. This outcome is coherent with the negative elevation trend occasionally reported in Figures S15 to S25.

Deriving a single blended lapse rate from Figure 6 was challenging, owing to the remarkable interannual variability in enhancement factors – especially above 3000 m ASL. Several options were considered, and a first assumption was made to exclude a indefinitely exponential growth such as that depicted by the dashed lines in Figure 6. We also assumed that enhancement factors larger than 6 were suspicious, and so we restricted the fitting pool to factors lower than 6. We finally postulated a maximum value of 3 for the fitted curve, supported by an evident cluster of snow-course-based enhancement factors from 2 to 3. While no consensus on this matter has been reached in the literature, these choices were based on scattered, but consistent pieces of evidence showing that precipitation gradients tend to saturate at very high elevation, once the bulk of orographic precipitation has been exhausted (e.g., see Alpert, 1986; Napoli et al., 2019). The resulting curve is depicted in red in Figure 6 and follows this equation:

$$\epsilon_f = 0.395 e^{0.628z}, \tag{1}$$

where $\epsilon_f$ is the predicted enhancement factor, $z$ is elevation in km, and $\epsilon_f$ is capped to 3 where Equation 1 exceeds 3. The 95% confidence bounds for the two parameters read (0.2857, 0.5041) and (0.5223, 0.7328), respectively, with coefficient of determination $r^2 = 0.44$ and Root Mean Square Error (RMSE) = 0.76.

Equation 1 was implemented in GRISO1 and used to correct predicted precipitation at virtual gauges above the precipitation-gauge line; this spatialization procedure was then re-run to take into account orographic gradients (GRISO2, see Section 3.2). We stress that Equation 1 only serves the scopes of this paper and is no definitive answer to the problem of capturing orographic gradients. More work should be dedicated to fully comprehend the large scatter in enhancement factors at high elevations and thus derive a more robust parametrization. Some starting points for future investigations are discussed in Section 5.

Snow-depth maps produced by including snow-course data in the calibration pool allocated on average more snowpack than those produced using only snow-depth sensors for all elevations and all water years at Beauregard (Figure 7, a to c). At Valpelline, the map obtained by including snow courses predicted on average more snow in 2017 and 2019 and less snow in 2018 than the snow-depth-sensor-based map (Figure 7, e to g). On average across all elevations and these three water years, the ratio between maps including snow courses and those using only snow-depth sensors was ∼ 1.5 at Beauregard and ∼ 2.1 at Valpelline, confirming a general underestimation obtained by using snow-depth sensors only. Snow-depth-sensor-based maps were multiplied by these ratios to remedy for this bias, as explained in Section 3.3.

Note that the bias of snow-depth maps was not caused by an underestimation of snow-depth lapse rates, as it could be expected based on results for precipitation (e.g., Figure 2). Instead, maps calibrated by including snow-course data predicted





steeper gradients than snow-depth sensors only in one case out of six (water year 2019 at Beauregard). This means that the relative underestimation of snowpack storage was often larger at the lowest elevations of the considered hydropower catchments (that is, ∼ 2000 m ASL), and progressively smaller at higher elevations.

## 4.4  Evaluation of water-balance predictive accuracy

Water-year cumulative precipitation obtained using only precipitation gauges (GRISO1, variable $P_{v1}$) was consistently smaller
than observed water-year cumulative streamflow (variable $Q_{obs}$) for all water years and both hydropower catchments (Figure 8). In particular, the ratio between $Q_{obs}$ and $P_{v1}$ (runoff coefficient) was between 1.56 and 1.77 at Beauregard and between 1.36 and 1.81 at Valpelline (Table 2). On the contrary, runoff coefficients using GRISO2 as informed by blended precipitation-gauge-snow-course lapse rates were between 0.78 and 0.85 at Beauregard and between 0.69 and 0.75 at Valpelline (Table 2). This means that, in contrast to GRISO1, GRISO2 predicted more annual precipitation (variable $P_{v2}$) that streamflow in both
catchments, which is generally expected in mountain catchments where subsurface storage plays a minor role in the annual water balance (see Section 5). Runoff coefficients using GRISO2 were also more consistent from year to year than those based on GRISO1 (Beauregard: standard deviation of 0.1 and 0.04 for GRISO1 and GRISO2, respectively; Valpelline: standard deviation of 0.25 and 0.03 for GRISO1 and GRISO2, respectively). This is a further piece of evidence that GRISO2 better captured precipitation patterns across the mountain landscape than GRISO1.

The underestimation of annual precipitation using GRISO1 was confirmed when looking at Full-Assim annual equivalent precipitation (Figure 8, variable $R_{v1}$ and $R_{v2}$ for GRISO1 and GRISO2, respectively); equivalent precipitation is the sum of rainfall, snowpack-, and glacier-runoff – Section 3.3. Values of $R_{v1}$ where closer to $Q_{obs}$ than $P_{v1}$, with ratios between $Q_{obs}$ and $R_{v1}$ between 1.03 and 1.24 at Beauregard and between 1.02 and 1.19 at Valpelline (Table 2). This confirms that assimilating snow maps in Flood-PROOFS successfully compensated for S3M's conceptual uncertainty in snow distribution
across the landscape (Boni et al., 2010), although some underestimation remained. Ratios between $Q_{obs}$ and $R_{v2}$ were, instead, consistently lower than 1 for all water years and both catchments (Table 2). Assimilating snow maps also reduced interannual variability in $Q_{obs}/R_{v1}$ compared to $Q_{obs}/P_{v1}$ (standard deviation of 0.11 and 0.09 at Beauregard and Valpelline, respectively), but $Q_{obs}/R_{v2}$ was still much more consistent from water year to water year than $Q_{obs}/R_{v1}$ (standard deviation of 0.04 and 0.07 at Beauregard and Valpelline, respectively).

Consistent with this general underestimation of incoming precipitation, simulated annual streamflow according to GRISO1 ($Q_{v1}$) was smaller than $Q_{obs}$ in four out of six catchment-water years ($Q_{v1}/Q_{obs}$ between 1.23 and 1.37 in Beauregard and between 0.95 and 1.08 in Valpelline – Table 2). Based on visual screening of observed vs. simulated cumulative hydrographs, the unexpected overestimation of $Q_{obs}$ by $Q_{v1}$ for two water years at Valpelline could be explained by an overestimation of late-summer runoff, likely because of underestimated evapotranspiration or overestimated convective rainfall (Figure 8, c and
g). In all other catchment-water years, underestimation started from the beginning of the water year and thus regarded both winter baseflow and summer peaks (Figure 8, b, f, j, and k). Although those two water years at Valpelline with more simulated than observed streamflow, biases of $Q_{v1}$ were negative for all water years and catchments and ranged from -150 to -10 mm (Table 2). This together with the comparatively high Root Mean Square Errors (RMSE, from ∼20 to ∼200 mm in Table 2)





confirmed that Flood-PROOFS simulations forced by GRISO1 generally underestimated water supply in these catchments.
Yet, Kling-Gupta Efficiencies (KGE, see Gupta et al., 2009; Kling et al., 2012) for $Q_{v1}$ were consistently higher than 0.65 and thus well above the benchmark represented by mean flow (-0.41, see Knoben et al., 2019), which has been often regarded as an arguable threshold between "bad" and "good" model performance (Schaefli and Gupta, 2007; Knoben et al., 2019). Contrary to RMSE and biases, KGE is the composition of bias, variability, shape, and timing error terms (Santos et al., 2018), meaning the issue with GRISO1-based simulations was really with total volume rather than with seasonal patterns.

Simulations using GRISO2 improved $Q_{v2}/Q_{obs}$ compared to $Q_{v1}/Q_{obs}$ for four out of six catchment-water years (Table 2 and Figure 8). Predictions of $Q_{v2}$ also yielded smaller biases and RMSEs (as absolute values) for all water years and catchments (Table 2). The improvement of $Q_{v2}$ over $Q_{v1}$ was particularly evident during the late-melt period (that is, from May on), when the highest elevations in these catchments start contributing runoff (Figure 8). Improvements during the accumulation period were much more modest, likely because streamflow generation during that period of the year is governed by processes that we
did not focus on here (e.g., groundwater flow, year-round glacier runoff due to basal melt). KGE coefficients also improved in five out of six catchment-water years, reaching values as high as 0.93.

Focusing on SWE, simulations of S3M using GRISO1 underestimated snow-course measurements both whether an Open-Loop and whether a Full-Assim mode was used (Figure 9). This underestimation was particularly significant for Open-Loop simulations, which agreed with results for precipitation vs. equivalent precipitation above (Table 2), whereas it showed no con-
sistent trend with elevation, which is instead consistent with results in Section 4.3 and Figure 7 regarding snowpack elevation gradients being particularly elusive to capture. Biases using GRISO2 were smaller than those using GRISO1 in five out of six catchment-water years (Full Assim mode), but again elevation trends were inconsistent (Figure 9). Using an Open-Loop simulations with GRISO2 would actually overestimate at the highest elevations in two out of six cases (2017 and 2018, Valpelline, Figure 9, d and h).
We derived two main results from this final focus on SWE: the first is an expected, net improvement in predicting high-elevation SWE when snow-course measurements are used in model development (especially at Beauregard). The second is that precipitation orographic gradients are highly seasonal as well as spatially variable, and remain challenging to fully capture with a one-fits-all approach as the one we used here (e.g., Equation 1 and Figure 7, d and h).

## 5  Discussion

### 5.1  Main findings

Snow courses have been a frequent option for conducting snow surveys since the seminal 1910 campaign by Church (1914, 1933) at Mount Rose, Nevada (US). Compared to stand-alone devices like snow pillows (Cox et al., 1978), courses allow operators to capture spatial variability of snow cover and so derive a more representative estimate of SWE across the landscape (Malek et al., 2017). This is why courses are now a cornerstone of water-supply forecasting in the western US (Pagano et al.,
2004; Harrison and Bales, 2016) and elsewhere (Metsämäki et al., 2005). In addition to their century-old role as indicator of snow water resources, in this paper we hypothesized that snow courses could be rethought as natural precipitation gauges,





in the hope that they could provide new information about precipitation totals and their orographic trends at elevations that are usually ungauged. This hypothesis follows intuitions by other authors, such as Lundquist et al. (2015) or Zhang et al. (2017), who used pillow SWE and snow depth as a surrogate of precipitation, respectively. Others, such as Immerzeel et al.

(2015), addressed this problem by inferring precipitation from glacier mass balances and runoff. Our novelty was to mine new information from snow courses, which provide spatial snapshots *in lieu* of point values.

The main findings of this paper in this regard are two. First, peak-season snow-course SWE above 3000 m ASL can be 2 to 8.5 times higher than measured winter cumulative precipitation at elevations below 2000 m ASL (Figure 6), with orographic trends that are up to five times those captured by the precipitation-gauge network (Figure 2). While orographic precipitation has

been a target of extensive research so far (see the Introduction), extrapolating precipitation-gauge signal above the precipitation-gauge line still lacks solid guidelines (Ruelland, 2020). In this paper, we contributed highly-needed, multi-year estimates of orographic trends across sharp altitudinal gradients.

Second, leveraging snow courses to refine the precipitation- and snow-depth-spatialization algorithms of an operational flood-forecasting chain (Flood-PROOFS) allowed for improvements in modeling accuracy not only for SWE (Figure 9), but

importantly for the whole water balance (Figure 8). This result is encouraging given that no model recalibration was performed, and as such we did not mix the effect of precipitation correction with other confounding factors that may lead to equifinality issues (Beven and Freer, 2001; Lundquist et al., 2015). Although the bulk of mountain river basins in the European Alps lies below 2000 m ASL (Elsen et al., 2020), areas above the precipitation-gauge line are a fundamental hydropower resource and represent a significant portion of higher-elevation mountain ranges such as the Himalayas. This paper outlined opportunities to

obtain more robust hydrologic predictions without necessarily investing into long recalibration efforts.

Despite these promising findings, the question remains whether our blending approach (Section 3.1) reconstructed the true precipitation lapse rate, or whether it captured other drivers of snowpack distribution at high elevations (e.g., wind drift). While both scenarios would be an argument in favor of using snow courses for informing hydrologic predictions, only the first would imply that we achieved the right answer for the right reason (Kirchner, 2006). In essence, this question points to determining

whether snow-course data are primarily a reflection of orographic precipitation or, e.g., of wind drift or solar radiation.

Several hints point to our reconstruction method capturing actual orographic trends in precipitation, rather than other snow-distribution processes. First, previous studies in the Alps already showed that annual mean precipitation at 1000-2000 m ASL is generally up to 2 times precipitation below 1000 m ASL, (Frei and Schär, 1998; Napoli et al., 2019), a mechanism that the precipitation network at Beauregard and Valpelline did not fully capture but that is consistent with our estimates of

blended orographic-enhancement factors (Figure 6 and Equation 1). Second, several previous attempts to improve estimates of hydrologic models in mountain regions through snow-data assimilation alone reported inconclusive results (Tang and Lettenmaier, 2010), whereas the clear improvements in this study suggest that we did capture at last some components of orographic precipitation in addition to snow patterns. Third, we computed orographic-enhancement factors by averaging snow-course measurements above 3000 m ASL rather than considering each of them individually, in an effort to reduce the effects of small-

scale spatial variability. Fourth, values of snow-course SWE at elevations close to the local precipitation-gauge line (∼2000 m ASL) are comparable to cumulative precipitation estimated by the nearest gauge (Figure 4), which suggests that the connection



between precipitation-gauge-based and snow-course-based lapse rates was smooth and realistic. Thus, we conclude that snow courses do bear a characteristic signature of precipitation orographic enhancement.

## 5.2 Implications

The fact that snow courses reflect orographic gradients has three main implications. First, it shows that the lack of measurements above the precipitation-gauge line plays an important role in the misrepresentation of orographic gradients. Many previous papers have already reported that precipitation-gauge networks tend to underestimate precipitation at high elevations, including Zhang et al. (2017) in the Sierra Nevada of California or Ruelland (2020) in the French Alps, among others. However, this misrepresentation has often been explained in combination with undercatch (Valery et al., 2009; Zhang et al., 2017; Collados-
Lara et al., 2018), rather than as a peculiar effect of precipitation undersampling at high elevations (Lundquist et al., 2015; Ruelland, 2020). In fact, the magnitude of orographic enhancement inferred through snow courses (Figure 6) was certainly larger than what could be estimated based on standard correction approaches for undercatch (see Rasmussen et al., 2012; Kochendorfer et al., 2017, and Section 2.2). Also, both GRISO1 and snow-depth-sensor-based maps underestimated these orographic gradients: we conclude that the main cause of this misrepresentation was the lack of data above the precipitation-
gauge line. Even though measuring precipitation across cryosphere-dominated headwaters is no easy task (Rasmussen et al., 2012), this paper shows that this is still an urgent priority for future research.

     Second, snow courses emerge as a valuable source of information not only to estimate snow water resources, but more generally precipitation distribution. This role of snow courses has been at the foundation of their use as predictors of annual runoff in water-supply forecasting (Hart and Gehrke, 1990; Pagano et al., 2004), but in that context snow courses are used in
a lumped way and their orographic signature is never fully leveraged. More in general, the value of snow courses has often been overlooked in favor of temporally more dense, but also much less spatially diverse automatic devices like snow pillows or snow-depth sensors. The result is that snow courses, or other labor-intensive survey methods, are a rare feature of operational snow surveys, when they have not been discontinued already (see the case of The Historical Snow Survey of Great Britain in Spencer et al., 2014). The significant variability in enhancement factors between Beauregard and Valpelline, along with the fact
that snow-course-based enhancement factors were significantly larger than those based on precipitation gauges, demonstrates that collecting spatially distributed snow data is still worth the effort. Moreover, the longest snow-course time-series will soon approach one century (Huning and AghaKouchak, 2020), meaning these courses could be used to explore orographic enhancement from a climatological standpoint. Recent advances in remote sensing, such as the Airborne Snow Observatory (Painter et al., 2016) or the Sentinel-1-based snow-depth retrieval algorithm by Lievens et al. (2019), could also be considered
as less time-consuming alternatives, with only a minor drop in accuracy – if any.

     Third, correctly capturing orographic gradients matters, despite this component being often simplified in many hydrologic models (see Ruelland, 2020, and the Introduction). More specifically, our results quantified that only relying on low- to mid-elevation precipitation gauges may lead to underestimating both headwater snowpack (Figure 9) and annual runoff (Table 2) by up to 50% or more. This agrees with Lundquist et al. (2015), who also found errors of 50% or more with respect to
high-elevation snow pillows for specific storms as represented by gridded precipitation datasets that only used low-elevation





precipitation data. Also, our results suggest that snow (and consequently runoff) predictions benefit from a multi-source frame-work, where both precipitation-spatialization and snow-assimilation protocols are involved (see again Figure 9).

### 5.3 Sources of uncertainty and outlook

We made a number of assumptions that may have represented sources of potential uncertainty and thus opportunities for future
work. First, we primarily focused on elevation gradients, since these are the most important factors driving the mountain water budget (Bales et al., 2006). Doing so was at the expenses of exploring other spatial patterns, such as those driven by aspect and slope. The variability in enhancement factors between the predominantly north-facing Beauregard and the south-to-west facing Valpelline suggests that these additional factors are important. Our results also showed significant interannual variability in enhancement factors, which somehow disagrees with the recurring finding that snow patterns are consistent from year to
year (Zheng et al., 2018). Future work could focus on deriving a multivariate alternative to Equation 1 that fully embraced physiographic features besides elevation, and in particular aspect, slope, and canopy cover (Malek et al., 2017). For example, Vögeli et al. (2016) showed significant improvements for a spatially distributed snow model when scaling its predictions with remote-sensing data. Winter climatology may also be considered as an additional predictor of orographic enhancement, for example in the form of predominant synoptic conditions as done by Garavaglia et al. (2010).

Second, we applied GRISO2 across the whole water year (September to August), although we estimated our orographic-enhancement factors only with winter data. This was done both for consistency reasons, and because we expect orographic enhancement to be at play during summer too. Nonetheless, mechanisms behind summer precipitation are significantly different from those behind winter precipitation (convective vs. stratiform, see e.g. Avanzi et al., 2015). No specific dataset can replace our snow courses during summer, but at the same time efficiency of precipitation gauges is much higher for rainfall than
snowfall (Peck, 1972). Thus, measuring precipitation at elevations above 3000 m ASL is much more feasible during summer than winter, which could be leveraged in the future to explore specific parametrizations of Equation 1 for summer.

Third, we converted snow-course snow depth to SWE using one average density value for each catchment-water year, with density being measured to a significantly smaller number of locations than snow depth due to logistical constraints. We expect this assumption to play a minor role in our assessment, given that snow-density spatial variability is much smaller than that of
snow depth (López Moreno et al., 2013). Yet, Raleigh and Small (2017) showed that snow-density modeling becomes the major source of uncertainty when mapping basin-wide SWE with Lidar. This means that targeted campaigns measuring snow-density variability across the landscape would still be beneficial to improve this work.

Fourth, peak-SWE date was assessed based on expert knowledge and changed from year to year. This protocol was different from other snow-course surveys, which are generally performed on recurring dates every year (Hart and Gehrke, 1990). Based
on results in Figure 5, we expect this assumption to have little to no impact on our estimates of orographic gradients. On the other hand, changing survey date based on when peak-SWE is supposed to occur favored our work because it allowed us to assume that snow-course SWE represented total winter precipitation falling at that location. Doing so would have been challenging with monthly courses performed on pre-defined dates.





Fifth, our assessment in Figure 8 assumed that the subsurface storage was a secondary source of annual streamflow compared
to precipitation, so that $Q/P < 1$ rather than $Q/P > 1$ was the desired outcome. We are lacking the necessary data to fully
resolve the water budget of these catchments, as done in California by Avanzi et al. (2020b). Still, the interannual consistency
in $Q/P$ based on GRISO2, together with the quasi linearity in precipitation-runoff relationship (Figure 3), do suggest a clear
correspondence between annual precipitation and annual runoff, and thus that $Q/P > 1$ by GRISO1 was suspicious.

Sixth, we did not use any additional precipitation dataset in addition to precipitation gauges (for example, radars). Besides
known issues with radars in mountain regions (see the Introduction), this was also because the present study leveraged the
precipitation-spatialization algorithm that is currently being maintained by the author team in Aosta Valley (Laiolo et al., 2014).
Future efforts by this team include an operational deployment of GRISO2 for flood forecasting, potentially in combination with
a newly developed, in-house precipitation product based on conditional merging between precipitation gauges and radar.

## 6 Conclusions

We addressed the recurring challenge of estimating precipitation across ungauged, high-elevation headwaters above the "precipitation-
gauge line". We did so by hypothesizing that snow courses could be rethought as natural precipitation gauges and thus be
leveraged to reconstruct blended precipitation-gauge-snow-course lapse rates. We found that winter precipitation estimated
through peak-SWE snow-course data was 2 to 8.5 times higher than what recorded by nearby precipitation gauges, evidence
that orographic precipitation in this inner Alpine valley develops with unexpectedly large orographic gradients that would be
missed by using low-to-mid-elevation precipitation gauges. These gradients were also miscaptured by snow-depth sensors,
both because their elevation range is only slightly larger than that of precipitation gauges and because they are installed in open
and flat areas. Elevation trends of snow-course data were highly seasonal, with only some correlation with mean seasonal snow
depth; this revealed a feedback mechanism between orographic enhancement and winter precipitation climatology that par-
tially challenged the generalization of our results. Blending precipitation-gauge and snow-course data into a unique lapse rate
and using this information in an operational hydrologic-modeling chain (Flood-PROOFS) allowed us to improve predictions
for not just Snow Water Equivalent, but importantly for the water budget (specifically, the interannual consistency of runoff
coefficients). Snow courses bear a signature of orographic enhancement, which we successfully mined to gain insight into the
process of orographic precipitation and improve real-world hydrologic models.

*Code and data availability.* Sources of data used in this are reported in Section 2.2, and comprise the Aosta Valley Regional Authority
(https://cf.regione.vda.it/portale_dati.php, visited on July 30, 2020), Compagnia Valdostana delle Acque (https://www.cvaspa.it/_welcome_/,
visited on July 30, 2020), and the Aosta Valley Environmental Protection Agency (http://www.arpa.vda.it/it/, visited on July 30, 2020). Flood-
PROOFS is described in Laiolo et al. (2014) and partially available at https://github.com/c-hydro (visited on July 30, 2020).





*Author contributions.* FA, GE, and SG conceived the investigation, with contributions from all coauthors. FA and GE carried out analyses and performed model simulations. EC, PP, GF, and UMdC collected snow-course data and prepared SWE maps, as well as shared general
585 knowledge about cryospheric processes in the study catchments. SR and HS provided weather and snow data collected by the Aosta Valley Regional Authority, as well as shared general knowledge about hydrologic processes in the study catchments. MC and SJ provided reconstructed streamflow data and shared general knowledge about hydropower processes in the study catchments. FA, GE, SG, and EC led result interpretations, with inputs from all coauthors. FA prepared the manuscript, with inputs from all coauthors.

*Competing interests.* Authors declare no competing interest.

590 *Acknowledgements.* This work was supported by the Aosta Valley Regional Administration and CVA S.p.A.





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





**Table 1.** Inventory of available snow-course surveys, including survey date (day/month), number of samples per survey (N), and mean survey elevation ($Z_{MEAN}$). Only a subset of the five areas was surveyed every year, due to budgetary and logistical constraints. The spatial distribution of these samples is showed in Figures S2-S12, Supporting Information. Note that 2641 out of 5349 measurements collected at Goillet in 2017 were performed on 10/04. Snow courses for this area and water year were part of a large intercomparison workshop, hence the much larger sample size than other years and areas.

| Water Year | Valpelline | | | Beauregard | | | Cignana | | | Gabiet | | | Goillet | | |
| --- | --- | --- | --- | --- | --- | --- | --- | --- | --- | --- | --- | --- | --- | --- | --- |
| | Date | N | $Z_{MEAN}$, m | Date | N | $Z_{MEAN}$, m | Date | N | $Z_{MEAN}$, m | Date | N | $Z_{MEAN}$, m | Date | N | $Z_{MEAN}$, m |
| 2008 | 24/06 | 519 | 3239 | - | - | - | - | - | - | - | - | - | - | - | - |
| 2009 | 25/05 | 1130 | 2932 | - | - | - | - | - | - | - | - | - | - | - | - |
| 2010 | 02/05 | 791 | 2882 | - | - | - | - | - | - | - | - | - | - | - | - |
| 2011 | 13/05 | 566 | 2932 | - | - | - | - | - | - | - | - | - | - | - | - |
| 2012 | 10/05 | 659 | 2826 | - | - | - | - | - | - | - | - | - | 10/05 | 95 | 2921 |
| 2013 | 16/04 | 1054 | 2847 | - | - | - | 16/04 | 257 | 2637 | - | - | - | 24/05 | 58 | 2832 |
| 2015 | 12/05 | 599 | 2931 | - | - | - | - | - | - | - | - | - | - | - | - |
| 2016 | 19/04 | 302 | 2936 | - | - | - | - | - | - | - | - | - | 19/04 | 226 | 2977 |
| 2017 | 10/04 | 738 | 2858 | 10/04 | 634 | 2544 | - | - | - | 10/04 | 382 | 2933 | 06/04 | 5349 | 2933 |
| 2018 | 11/05 | 1124 | 2784 | 10/05 | 572 | 2433 | 24/04 | 593 | 2683 | 21/04 | 338 | 3004 | - | - | - |
| 2019 | 17/04 | 755 | 2922 | 30/04 | 853 | 2627 | 18/04 | 257 | 2643 | 11/04 | 248 | 3004 | 18/04 | 193 | 3059 |





**Table 2.** Evaluation metrics of Flood-PROOFS simulations driven by GRISO1 vs. those driven by GRISO2. The first distributed precipitation only using precipitation gauges, whereas the second included an orographic correction developed in the present study based on snow courses. $Q$, $P$ and $R$ are annual streamflow, precipitation, and equivalent precipitation, respectively (with equivalent precipitation being the sum of rainfall, snowpack runoff, and glacier runoff). $obs$, $v1$ and $v2$ refers to observed data and simulated data according to GRISO1 and GRISO2, respectively. RMSE is Root Mean Square Error, bias is simulated minus observed, and KGE is the Kling-Gupta efficiency according to Kling et al. (2012).

| Metric | Beauregard | | | Valpelline | | |
|---|---|---|---|---|---|---|
| | 2017 | 2018 | 2019 | 2017 | 2018 | 2019 |
| Precipitation $P$ | | | | | | |
| $Q_{obs}/P_{v1}$ (-) | 1.56 | 1.69 | 1.77 | 1.39 | 1.36 | 1.81 |
| $Q_{obs}/P_{v2}$ (-) | 0.78 | 0.86 | 0.85 | 0.69 | 0.74 | 0.75 |
| Equivalent precipitation $R$ | | | | | | |
| $Q_{obs}/R_{v1}$ (-) | 1.03 | 1.24 | 1.06 | 1.02 | 1.04 | 1.19 |
| $Q_{obs}/R_{v2}$ (-) | 0.75 | 0.83 | 0.8 | 0.71 | 0.75 | 0.85 |
| Reconstructed streamflow $Q$ | | | | | | |
| $Q_{obs}/Q_{v1}$ (-) | 1.23 | 1.37 | 1.28 | 0.95 | 0.95 | 1.08 |
| $Q_{obs}/Q_{v2}$ (-) | 0.93 | 0.96 | 0.95 | 0.91 | 0.86 | 1.00 |
| $RMSE_{v1}$ (mm) | 114.99 | 201.72 | 145.69 | 19.52 | 42.87 | 74.38 |
| $RMSE_{v2}$ (mm) | 31.93 | 47.07 | 53.85 | 44.03 | 86.02 | 62.52 |
| $Bias_{v1}$ (mm) | -83.05 | -149.74 | -116.96 | -10.5 | -19.25 | -61.47 |
| $Bias_{v2}$ (mm) | 24.41 | -30.79 | -16.38 | 5.63 | 18.78 | -42.34 |
| $KGE_{v1}$ (-) | 0.79 | 0.65 | 0.71 | 0.95 | 0.86 | 0.82 |
| $KGE_{v2}$ (-) | 0.93 | 0.88 | 0.88 | 0.89 | 0.87 | 0.81 |

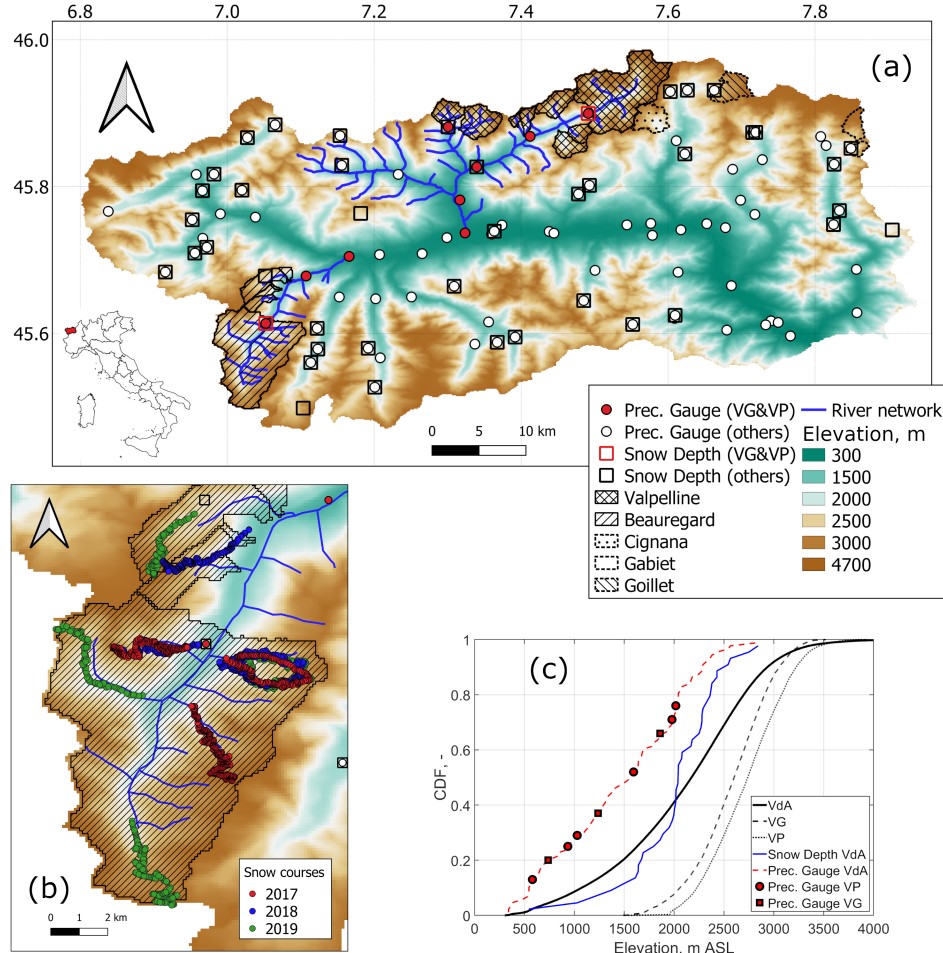

**Figure 1.** Panel (a): topography of Aosta Valley (VdA, our focus region), along with hydrography and hydropower-catchment delineation of the two valleys for which we reconstructed blended precipitation lapse rates (VG is Beauregard and VP is Valpelline). This panel also reports location of all precipitation gauges and snow depth sensors available to the present study, as well as catchment delineation of three other hydropower systems in Aosta Valley where snow courses were collected (see Table 1). Panel (b) shows examples of snow-course locations for three water years at Beauregard. Panel (c) reports elevation distribution of Aosta Valley, Beauregard and Valpelline, and of the precipitation-gauge and snow-depth-sensor networks.



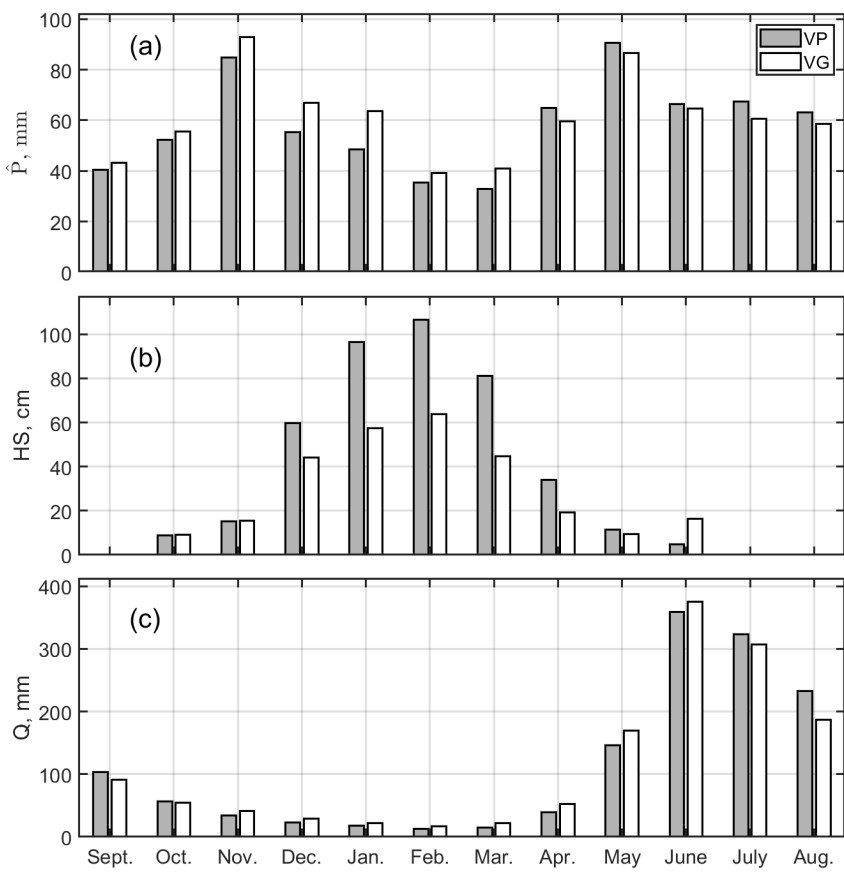

**Figure 2.** Average monthly cumulative precipitation (a), snow depth (b), and cumulative reconstructed streamflow (c) for Beauregard and Valpelline. Precipitation was calculated across all gauges in the valleys of Beauregard and Valpelline (variable $\hat{P}$, see gauge locations in Figure 1). Snow depth was from two representative, mid-elevation sensors (elevation was $\sim$1860 and 1970 m ASL for the snow depth sensor of Beauregard and Valpelline, respectively). Reconstructed streamflow refers to the hydropower catchments delineated in Figure 1. The reference period used for these statistics varied from variable to variable due to data gaps (see Section 4.1 for details). VG and VP are Beauregard and Valpelline, respectively.

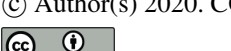



**Figure 3.** Summary of daily precipitation (a), snow depth (b), and reconstructed streamflow (c) for Valpelline. Precipitation was calculated across all gauges in the valleys of Valpelline (variable $\hat{P}$, see gauge locations in Figure 1). Snow depth was from a representative, mid-elevation sensor (elevation was ∼1970 m ASL). Reconstructed streamflow refers to the hydropower catchment delineated in Figure 1. Low, medium, and high snow seasons were estimated based on percentiles of mean seasonal snow depth at Beauregard, which had a complete record between water years 2008 and 2019 (Figure S14). This classification is only functional to the scopes of this paper and has no long-term climatological meaning. Panel (d) reports the estimated precipitation-runoff relationship for both hydropower catchments, where WY $\hat{P}$ and WY $Q$ are water-year cumulative precipitation and reconstructed streamflow, respectively ($\hat{P}$ is systematically smaller than $Q$ because high-elevation headwaters are ungauged).





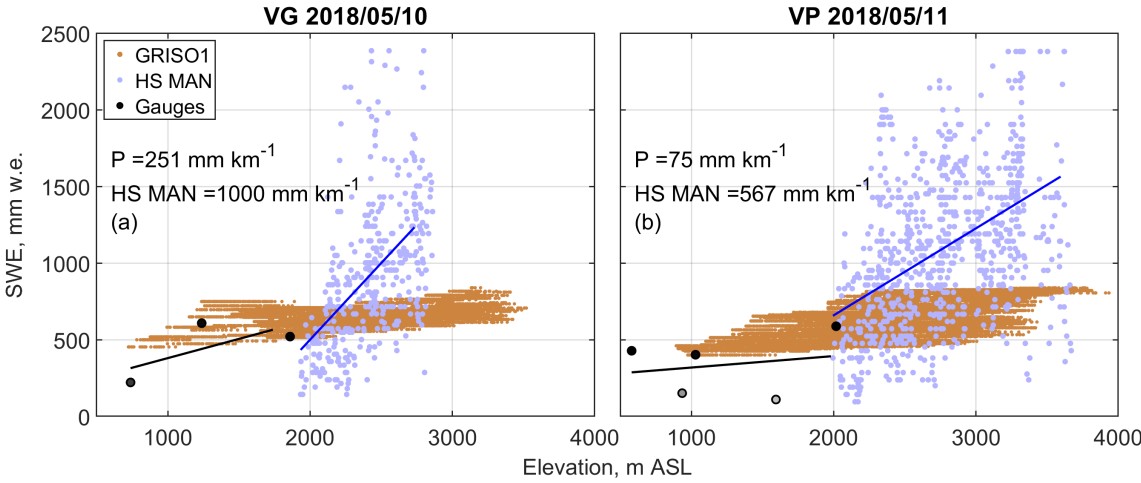

**Figure 4.** Winter orographic gradients estimated by precipitation gauges (black dots and line) vs. those estimated by snow courses (blue dots and line, HS MAN) in 2018. The grey scale for precipitation-gauge data points measures the amount of gaps in the time series, with black meaning a complete time series and white a time series with nearly 100% missing data points. The brown cloud is the winter orographic gradient estimated by GRISO1, which only relied on precipitation gauges. VG and VP are Beauregard and Valpelline, respectively. Similar plots for other water years are reported in the Supporting Information, Figures S15 to S25.



**Figure 5.** Annual winter orographic gradients estimated by precipitation gauges, snow courses (HS MAN), and the two snow maps assimilated by Flood-PROOFS at Valpelline (a) and Beauregard (b). HS map and SWE map are the snow-depth-sensor map and the SWE map, respectively. Details on Flood-PROOFS and these assimilated maps are reported in Section 3.3. Panels (c) and (d) show estimated orographic gradients by snow courses as a function of mean course snow depth above 3000 m ASL (Avg. HS MAN) and survey day of the year (DOY), respectively. $\rho$ is Pearson's correlation coefficient.





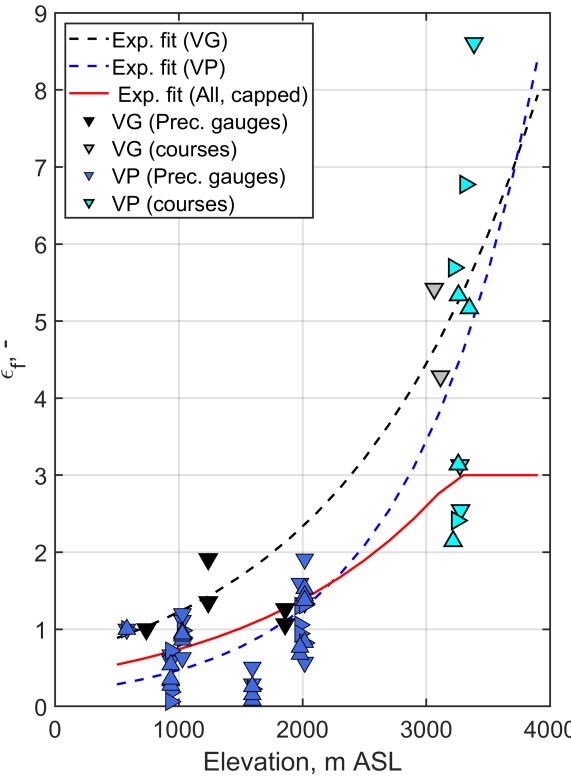

**Figure 6.** Orographic-enhancement factors $\epsilon_f$ for Beauregard (VG) and Valpelline (VP) across all water years as estimated using precipitation gauges (blue and black) and snow courses (grey and light blue). The dashed lines are exponential fits between orographic-enhancement factors and elevation, while the red line is a capped-exponential fit that was chosen to be implemented in Flood-PROOFS (Equation 1). Details on this choice are reported in Section 4.3, while details on Flood-PROOFS are reported in Section 3. Enhancement factors are the ratio between winter cumulative precipitation measured by gauges or estimated through snow courses above 3000 m ASL and precipitation measured by the lowest-elevation gauge in the same valley. The orientation of triangles responds to the classification of each water year in terms of mean snow depth, see Figure 3.

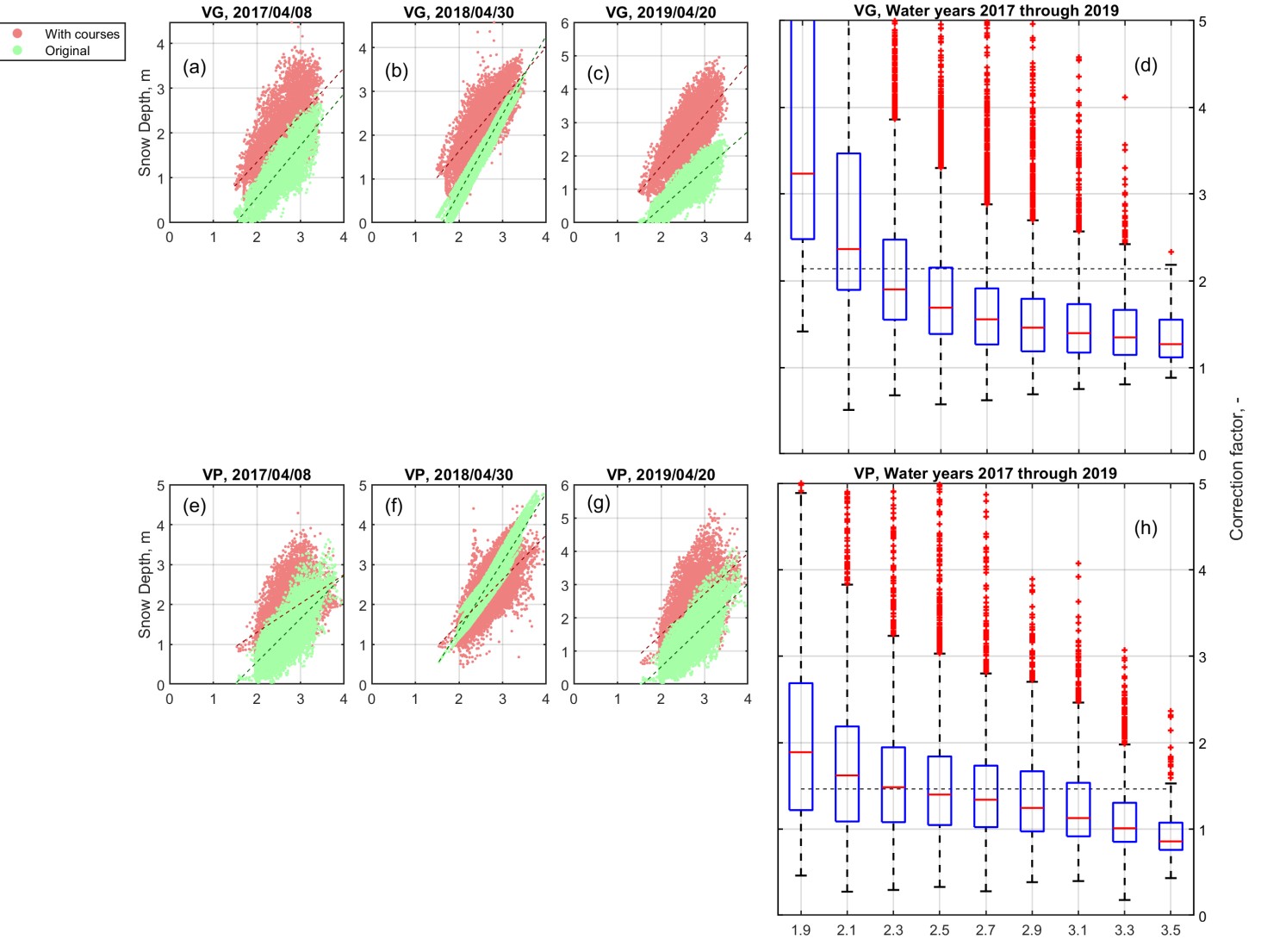

**Figure 7.** Orographic gradients of snow-depth maps derived by only using snow-depth sensors vs. those derived including high-elevation snow courses in the calibration pool (panels a to c for Beauregard and panels e to g for Valpelline). Panels d and h show the orographic trend of the ratio between maps derived including snow courses in the calibration pool and those derived by only using snow-depth sensors. VG and VP are Beauregard and Valpelline, respectively.







**Figure 8.** Water-balance evaluation of Flood-PROOFS simulations driven by GRISO1 vs. those driven by GRISO2. Panels a, e, and i report a comparison at Beauregard (VG) between annual precipitation ($P$), equivalent precipitation ($R$), and streamflow ($Q$) according to GRISO1 and GRISO2 ($v1$ and $v2$, respectively). Panels d, h, and l report a similar comparison at Valpelline (VP). Panels b-c, f-g, and j-k compare observed daily cumulative reconstructed streamflow with simulated streamflow using GRISO1 or GRISO2. Equivalent precipitation is the sum of rainfall, snowpack runoff, and glacier runoff. Simulations were carried out in Full-Assim mode (see Section 3.3).



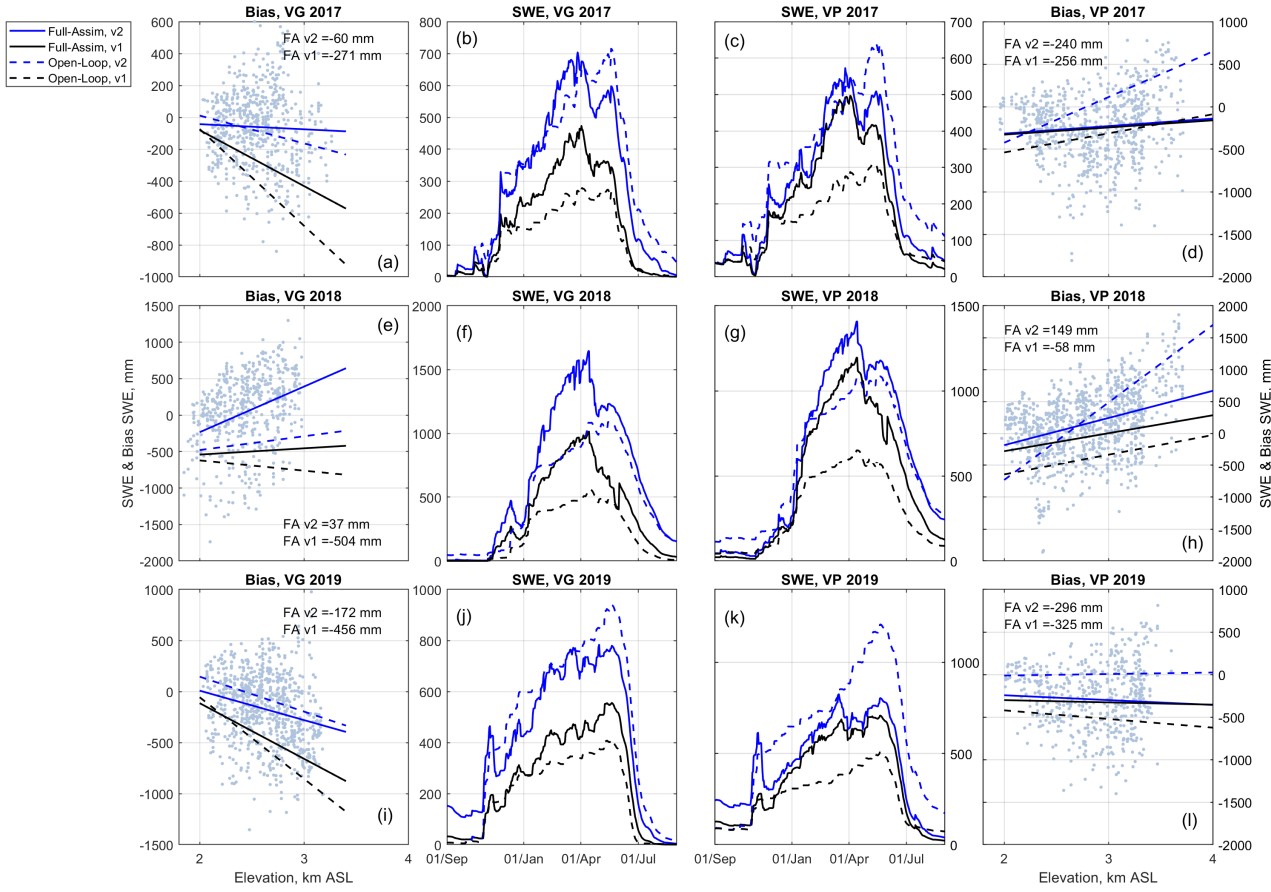

**Figure 9.** Evaluation of Flood-PROOFS simulations of Snow Water Equivalent (SWE) driven by GRISO1 vs. those driven by GRISO2 ($v1$ and $v2$, respectively). Panels a, e, and i report orographic trends of model bias with respect to snow-course measured SWE at Beauregard (VG). Panels d, h, and l report a similar comparison at Valpelline (VP). Panels b-c, f-g, and j-k compare simulated SWE using both GRISO1 or GRISO2. Simulations were carried out both in Open-Loop and in Full-Assim mode (see Section 3.3).