# Peer review of "Learning about precipitation lapse rates from snow-course data improves water-balance modeling"

_Hydrology and Earth System Sciences, 2020_

## Referee Comment (RC1) · Anonymous Referee #1 · 10 Jan 2021

The study introduces a way of implementing snow course data to get a better estimate of precipitation gradients in high elevations. Therefore the authors hypothesize the snow course data to serve as additional precipitation gauges (totalisators) and test this with runoff ratios as well as in the performance of predictions of a snow-hydrologic modelling chain. The paper is written clear and is well structured. I have only some minor comments to be clarified before I recommend publication.

- snow course representatively: 1. Are there concave features in the snow courses that influence snow depth and as consequence the calculated lapse rates and if are these representative for the hypsography of the catchment(s) or could they introduce

a bias in the estimation? If that is the case how could that be accounted for? Please add some words on this issue in the discussion. 2. How about wind drift effects in the snow courses? Do the courses under- or overcatch or do the authors think that wind drift is covered well with the courses used (considering also that wind drift might vary depending on the weather pattern)

- ephemeral snowpack: how are these accounted for in the calculation of the elevation gradients? If the precipitation reaches the ground and infiltrates the assumption of the snowpack as totalizator does not hold anymore

- glacier melt: the authors mention that the catchments are influenced by glaciers. Please add some information on how much the melt water might influence the observed streamflow

- application of lapse rate for summer period: The authors discuss that their use of the estimated lapse rate also in summer (full year simulation) is not optimal. I see that this is problematic particularly because of the different dominant precipitation type during summer. Can this not be disentangled in the interpretation or can this in the simulation not be changed in the first place?

Please, find the line by line comments below.

Minor comments: L27 the impact on societies is not obvious, please add a short example or a better explanation here

L120 filtering regarding which aspects? please add

L125-127 please add based on what (which evaluation) that was found to be best

L297 ephemeral snowpack: how are these accounted for in the calculation of the elevation gradients? If the precipitation reaches the ground and infiltrates the assumption of the snowpack as totalizator does not hold anymore

editorial comments: L2 switch on and both
L6 add "of" after "upstream"

L10 remove "quite"

L13 "and importantly precipitation vs. observed streamflow" not clear does that belong to remedy still?

L14 "to year that" -> " to year than"

L23 add "to" after "contribute"

L24 Is there a "as" missing at the beginning of the sentence?

L28 add "add" before "better and "of" before "how"

L37 showed -> shown

L82 then -> secondly

L128 add "the" before "measurement"

L148 add "the" after "from"

L157 add "the" before "location"

L160 add "the" before "derive"

L194 necessarily -> expected to be

L217 dynamical -> dynamic

L257 remove "will"

L262 add "the" before "water years"

L264 add "the" before "original"

L294 with -> to

L305 add "during winter" after "higher"

L306 "in Valpelline during winter and summer, respectively" -> "during summer in Valpelline"

L306 "is favor" -> "in favor"

L312 remove "these two"

L317 "peak SWE date occurs" -> "SWE peaks"

L460 remove "the" after "see"

L509 here you could refer to innovative measurement developments that make the snow course measurement much more easier and effective such as the study by Griessinger et al. 2018

References Griessinger, N, Mohr, F, Jonas, T. Measuring snow ablation rates in alpine terrain with a mobile multioffset ground‐penetrating radar system. Hydrological Processes. 2018; 32: 3272– 3282. https://doi.org/10.1002/hyp.13259

---

## Referee Comment (RC2) · Anonymous Referee #2 · 8 Feb 2021

The paper introduces a method to combine snow-course data with precipitation gauges to improve the water balance modelling in complex terrain. The mapper is very well written and the results are well presented. Please find minor comments listed below: Title: I would suggest changing the title as I think the snow course data are used to get more information on lapse rates of precipitation affected by a lot of different processes at the mountain and the ridge scale rather than only orographic enhancement.

Introduction: the spatial variability of precipitation and in particular of snow can be caused by different processes acting at different scales. At mountain to ridge scales orographic enhancement but also the effect of preferential deposition of precipitation

can drive the spatial distribution of precipitation and can also have large effects on the snow course measurements as well as snow gauges. I would ask the authors to shortly add a discussion on that to the Introduction part as many previous studies could show that preferential deposition of solid precipitation might have strong effects on snow distribution at high elevations (.e.g Gerber et al., 2017; Gerber et al., 2019).

L 156: please provide some details on the typical location of those snow courses - are those similar to snow stations typically located at wind-sheltered locations? Please also provide more details how the transects f such snow courses were selected. This might have an important effect on the representative of such snow courses.

Figure 3 – how did you classify between low snow medium snow and high snow. Does low snow class also include ephemeral snowpack?

L 180: there are studies such as Grünewald et al., 2014 or Colladon-Lara et al., 2018 who showed a decrease in snow height at very high elevations - i.e. inverse trend above a certain elevation. Did you also account for that? This might have a strong effect on your factors if using elevations above 3000 m ASL as natural precipitation gauge.

Figure 7: no colour blind-figures are used.

L 195: as convection driven storms will totally change precipitation distribution I would suggest only using peak-season SWE measurements for solid precipitation

Could you elaborate on measurement accuracy of precipitation gauges in case of solid precipitation (i.e. wind drift on falling snow flakes)

L 229: I not fully understand why at this point the elevation threshold of 2700 m is used

L 306: in favour L 475: please list also preferential deposition of snowfall which might have an effect on your measurements Suggested references: Gerber, F., Mott, R., & Lehning, M. (2019). The Importance of Near-Surface Winter Precipitation Processes in Complex Alpine Terrain, Journal of Hydrometeorology, 20(2), 177-196. Gerber, F.,

Lehning, M., Hoch, S. W., and Mott, R. (2017), A close‐ridge small‐scale atmospheric flow field and its influence on snow accumulation, J. Geophys. Res. Atmos., 122, 7737– 7754, doi:10.1002/2016JD026258. Grünewald, T., Bühler, Y. and Lehning, M. (2014) Elevation dependency of mountain snow depth. The Cryosphere, 8, 2381– 2394. https://doi.org/10.5194/tc-8-2381-2014. Collados‐Lara, A‐J, Pardo‐Igúzquiza, E, Pulido‐Velazquez, D, Jiménez‐Sánchez, J. Precipitation fields in an alpine Mediterranean catchment: Inversion of precipitation gradient with elevation or undercatch of snowfall? Int J Climatol. 2018; 38: 3565– 3578. https://doi.org/10.1002/joc.5517
* * *

---

## Editor Comment (EC1) · Bettina Schaefli (Editor) · 9 Feb 2021

The paper has received two reviews, which are both very positive and make only minor suggestions to further improve the manuscript. I invite the authors to answer the questions / comments in the public discussion before preparing the revised version.

---

## Author Response (AR1)

Savona (Italy)

March 4, 2021

Dear Prof. Bettina Schaefli, Editor,

We would like to submit the manuscript *Learning about precipitation lapse rates from snow-course data improves water-balance modeling* for publication in HESS. The manuscript is a resubmission of manuscript **hess-2020-571**, which was reviewed by two referees.

We have extensively revised the manuscript based on comments from both referees and would like to thank all of you for finding the time to review our manuscript. We confirm that all requested changes were feasible and we welcomed all of them.

Please find attached our point-by-point replies and the new version of our manuscript for details. We also attached a version of the manuscript with tracked changes.

With our best regards,

*Francesco Avanzi and coauthors*

**Reply to Referee #1**

**The study introduces a way of implementing snow course data to get a better estimate of precipitation gradients in high elevations. Therefore the authors hypothesize the snow course data to serve as additional precipitation gauges (totalisators) and test this with runoff ratios as well as in the performance of predictions of a snow-hydrologic modelling chain. The paper is written clear and is well structured. I have only some minor comments to be clarified before I recommend publication.**

> Public response: We thanks Reviewer #1 for their constructive comments. We confirm that all requested revisions are feasible and we will work in this direction as soon as the interactive discussion will be finalized.

**Snow course representatively: 1. Are there concave features in the snow courses that influence snow depth and as consequence the calculated lapse rates and if are these representative for the hypsography of the catchment(s) or could they introduce a bias in the estimation? If that is the case how could that be accounted for? Please add some words on this issue in the discussion.**

> Public response: We confirm that topographic patterns in our study region are particularly complex, including an alternation of convex-concave features. However, the spatial scale of the process we investigate here (orographic enhancement) is certainly much larger than that involved in snow deposition in concave features and snow erosion in convex features. This together with our choice of spatially averaging snow-course data above 3000 m ASL, rather than considering each data point, aimed at minimizing the impact of such local effects on our estimates of precipitation gradients. We agree that it is worth commenting on this matter in the Discussion and will do so.
>
> Changes to the manuscript: We added a passage on this matter both in the Introduction and in the Discussion section (see lines **77ff** and **555ff**).

**2. How about wind drift effects in the snow courses? Do the courses under- or overcatch or do the authors think that wind drift is covered well with the courses used (considering also that wind drift might vary depending on the weather pattern)**

> Public response: We agree with the reviewer that wind drift is certainly a driver of snow distribution at high elevations in our study region. It is also well known that wind reduces SWE at high elevations through sublimation of blowing snow. Our choice of spatially averaging snow-course data above 3000 m ASL, rather than considering each data point, aimed at minimizing the impact of such local effects on our estimates of precipitation gradients (see also the previous comment regarding concave features). Also, the sampling protocol avoided known deposition or erosion areas, as far as this was possible. Both aspects increase our confidence that the large-scale precipitation gradients presented in this paper are only marginally impacted by wind-drift effects. Still, we acknowledge that this is certainly a factor to consider and we will expand the discussion to touch on this.
>
> Changes to the manuscript: We added a passage on this matter in the Introduction, in the Data, and in the Discussion section (see lines **77ff**, **169**, and **555ff**).

**- ephemeral snowpack: how are these accounted for in the calculation of the elevation gradients? If the precipitation reaches the ground and infiltrates the assumption of the snowpack as totalizator does not hold anymore**

Public response: Ephemeral snowpacks would indeed challenge the overarching assumption of our orographic-gradient estimation method, as this is based on peak SWE being a direct measure of total precipitation during the snow season. We were already mentioning this at lines 178ff page 6, where we discussed that such instances are relatively rare at the investigated elevations above 3000 m ASL. In this regard, we also pointed out that we defined the onset of the snow season from the first hour with at least 20 cm of snow on the ground for a reference snow-depth sensor (see the original manuscript). This aimed at capturing precipitation totals for the bulk of the accumulation season, while excluding early-season snowfall events that might result in complete or partial depletion of the snowpack. We appreciate this comment and we will be more explicit on this matter in the revised manuscript.

Changes to the manuscript: We clarified the passage on this matter in the Methods (see lines **197ff**).

**- glacier melt: the authors mention that the catchments are influenced by glaciers. Please add some information on how much the melt water might influence the observed streamflow**

Public response: We confirm that both catchments are partially covered by glaciers, the contribution of which to total streamflow is challenging to assess due to a lack of measurements. Our models may provide a quantification of this contribution, but we preferred not to include this in the paper given that our glacier implementation in these specific valleys has never been fully validated (again, due to the lack of measurements). Based on qualitative inspection of the observed hydrographs, we expect glaciers to particularly contribute to late-summer streamflow, when input from snowmelt declines. This is in line with glacier role in other catchments across the Alps.

Changes to the manuscript: We added some information on this matter in the Data section (see lines **103ff**).

**- application of lapse rate for summer period: The authors discuss that their use of the estimated lapse rate also in summer (full year simulation) is not optimal. I see that this is problematic particularly because of the different dominant precipitation type during summer. Can this not be disentangled in the interpretation or can this in the simulation not be changed in the first place?**

Public response: We agree that this is an open issue for future work. On the one hand, one may hypothesize that orographic enhancement exists both for stratiform and convective precipitation, and especially during early fall or spring the latter may contribute to some extent to peak-SWE values as the former. However, synopctic-scale circulation and its inter-actions with mountains are significantly different between the winter and the summer season. While one may compare simulations with or without summer orographic enhancement to draw some preliminary conclusions on this matter, it is also true that doing so would leave several questions unanswered and potentially raise further issues. For example, is this difference really due to orographic precipitation, or is it related to how Flood-PROOFS parametrizes evapotranspiration? May snowmelt infiltration and so groundwater recharge also play a role?

In practice, we are currently developing an operational version of this algorithm, where

we are considering whether this orographic-enhancement spatialization approach should be limited to winter only. In this paper, we preferred to apply it to the entire water year both for consistency and because we deemed that an exhaustive discussion of winter vs. summer precipitation patterns and their relation with orography would go well beyond the scope and brevity of one paper. Doing so may also add confusion and dilute the core message. We will improve our discussion based on the points above, including some operational outlooks.

Changes to the manuscript: We added some information on this matter in the Discussion (see lines **568ff**).

Public response: *All minor comments will be addressed in the revised manuscript. Here, we comment on those requiring further details from our side.*

**L27 the impact on societies is not obvious, please add a short example or a better explanation here**

Public response: A very simple example in this regard is that the wet side of continental orographic barriers has historically been much more populated than the dry side, which often corresponds to deserts (e.g., the Atacama desert or to some extent the California eastern Sierra region). In Europe, this has corresponded to different timing and amount of ecosystem services such as the seasonal freshet. We will improve this passage.

Changes to the manuscript: Adding the needed context would have compromised conciseness here, so this unnecessary reference to societies was removed (see lines **28ff**).

**L120 filtering regarding which aspects? please add**

Public response: Filters include out-of-range or negative values (where applicable, for example for snow depth). That said, the main strength of this dataset is the supervised-filtering part, with one expert visually screening weather data on a periodical basis (roughly every week, although this varies) and assigning quality flags to each data point.

Changes to the manuscript: We added the above information in the Data section (see lines **128ff**).

**L125-127 please add based on what (which evaluation) that was found to be best**

Public response: The evaluation involved comparing precipitation totals at snow-depth sensor locations with concurrent snow-depth increases. Precipitation totals were estimated using various parametrizations and that by Allerup et al. (1997) was found to yield the lowest error (unpublished work).

Changes to the manuscript: We added the above information in the Data section (see lines **134ff**).

**L13 "and importantly precipitation vs. observed streamflow" not clear does that belong to remedy still?**

Public response: Yes, we will fix this.
Changes to the manuscript: This passage was fixed (see line **13**).

**L509 here you could refer to innovative measurement developments that make the snow course measurement much more easier and effective such as the study by Griessinger et al.**

**2018**

Public response: Agreed, we will include this.
Changes to the manuscript: Added (see line **533**).

**Reply to Reviewer #2**

**The paper introduces a method to combine snow-course data with precipitation gauges to improve the water balance modelling in complex terrain. The mapper is very well written and the results are well presented. Please find minor comments listed below.**

> Public response: We thanks Reviewer #2 for their constructive comments. We confirm that all requested revisions are feasible and we will work in this direction as soon as the interactive discussion will be finalized.

**Title: I would suggest changing the title as I think the snow course data are used to get more information on lapse rates of precipitation affected by a lot of different processes at the mountain and the ridge scale rather than only orographic enhancement.**

> Public response: Agreed. The new title will be *"Learning about precipitation **lapse rates** from snow-course data improves water-balance modeling*
> Changes to the manuscript: Title changed as recommended.

**Introduction: the spatial variability of precipitation and in particular of snow can be caused by different processes acting at different scales. At mountain to ridge scales orographic enhancement but also the effect of preferential deposition of precipitation can drive the spatial distribution of precipitation and can also have large effects on the snow course measurements as well as snow gauges. I would ask the authors to shortly add a discussion on that to the Introduction part as many previous studies could show that preferential deposition of solid precipitation might have strong effects on snow distribution at high elevations (.e.g Gerber et al., 2017; Gerber et al., 2019).**

> Public response: Agreed. This concept will be briefly mentioned in the Introduction.
> Changes to the manuscript: We added this point to the Introduction and the Discussion sections (see lines **77ff** and **490ff**).

**L 156: please provide some details on the typical location of those snow courses - are those similar to snow stations typically located at wind-sheltered locations? Please also provide more details how the transects f such snow courses were selected. This might have an important effect on the representative of such snow courses.**

> Public response: Snow courses are not snow stations, they are a snow-survey protocol: snow depth is manually measured every 50 to 100 m along transects of several kilometers (see line 146ff in the manuscript). This protocol aims at capturing snow-depth distribution in a way that is more representative of the landscape than stand-alone stations like ultrasonic depth sensors, which instead tend to overestimate both peak SWE and the duration of the snow season (Malek et al., 2017). In the present study, another asset of snow courses is that they captured the orographic gradient in snow depth (and so SWE), as they were collected from the local snow line up to the catchment divide (see Figure 1(b) in the manuscript for some examples).
> The term *snow course* is widely used in areas of the world where water-supply forecasting decisively depends on snow, such as the western US (Rice and Bales, 2010) or Finland (Lundberg and Koivusalo, 2003) – also see `https://www.wcc.nrcs.usda.gov/factpub/sect_4a.html`. We will add details above in the manuscript (line 146ff).

    Changes to the manuscript: We added some of the above details to the Introduction (see lines **68ff**).

**Figure 3 – how did you classify between low snow medium snow and high snow. Does low snow class also include ephemeral snowpack?**

    Public response: Low-, medium-, and high-snow water years were estimated based on percentiles of mean seasonal snow depth at Beauregard (see caption of Figure 3). Any water year with mean seasonal snow depth below the 33° percentile was classified as low-snow water year. Likewise, medium-snow water years had mean seasonal snow depth between the 66° and the 33° percentiles, with high-snow water years having mean seasonal snow depth above the 66° percentile. Ephemeral-snow water years were not attributed to any of these classes *a priori*, since this attribution depends on the magnitude of mean seasonal snow depth. Yet, it is likely that ephemeral-snow water years also have a low mean seasonal snow depth. We will add this in the manuscript.

    Changes to the manuscript: We added this information to the caption of Figure 3.

**L 180: there are studies such as Grünewald et al., 2014 or Colladon-Lara et al., 2018 who showed a decrease in snow height at very high elevations - i.e. inverse trend above a certain elevation. Did you also account for that? This might have a strong effect on your factors if using elevations above 3000 m ASL as natural precipitation gauge.**

    Public response: Good point! This decrease in snow height for very high elevations may be the result of various processes, including exhaustion of orographic-precipitation effects (Napoli et al., 2019), an increase in snow sublimation due to strong winds, or more generally interactions between high-elevation steep topography and snow redistribution processes (i.e., wind erosion, avalanches). Because we spatially averaged snow-course data above 3000 m ASL, rather than considering each data point, such multilinear trends in SWE at very high elevations were not explicitly modeled, but only implicitly embedded in our estimates of precipitation lapse rates. While one may consider spatially averaging snow-course data across smaller elevation bands to capture such multilinear patterns, we argue that such small-scale gradients based on snow courses may be counfounded by other processes, such as snow deposition in concave features and snow erosion in convex features. Because our predictions of the water balance dramatically improved even by using only spatially averaged snow-course data above 3000 m ASL, we conclude that multilinearity in lapse rates for very high elevations is likely a second-order effect in mountain hydrology compared to orographic enhancement across elevation gradients of various kilometers. We will add this discussion to the manuscript.

    Changes to the manuscript: We added this point to the Discussion section (see lines **555ff**).

**Figure 7: no colour blind-figures are used.**

    Public response: Agreed. We will improve Figure 7.
    Changes to the manuscript: Fixed.

**L 195: as convection driven storms will totally change precipitation distribution I would suggest only using peak-season SWE measurements for solid precipitation**

Public response: This was indeed the intended meaning of that sentence. The implicit hypothesis here was that liquid precipitation during winter above 3000 m ASL is negligible. We will clarify that passage.

Changes to the manuscript: We clarified this passage in the Methods section (see lines **211ff**).

**Could you elaborate on measurement accuracy of precipitation gauges in case of solid precipitation (i.e. wind drift on falling snow flakes)**

Public response: The evaluation of measurement accuracy of precipitation gauges involved comparing precipitation totals at snow-depth sensor locations with concurrent snow-depth increases. So the stated accuracy (see lines 125ff in the manuscript) is actually more representative of solid than liquid precipitation. We will clarify this in the manuscript.

Changes to the manuscript: We clarified this point (see lines **138ff**).

**L 229: I not fully understand why at this point the elevation threshold of 2700 m is used**

Public response: 2700 m ASL represents the "precipitation-gauge line" in this region, that is, the elevation above which no precipitation gauge is located (see the Introdution and Figure 1(c)). We will clarify this at line 229ff.

Changes to the manuscript: We clarified this point (see lines **244ff**).

**L 306: in favour L 475: please list also preferential deposition of snowfall which might have an effect on your measurements**

Public response: Agreed.

Changes to the manuscript: Point added (see line **490**).

**References**

Allerup, P., Madsen, H., Vejen, F., 1997. A Comprehensive Model for Correcting Point Precipitation. Hydrology Research 28, 1–20. URL: `https://doi.org/10.2166/nh.1997.0001`, doi:`10.2166/nh.1997.0001`, arXiv:`https://iwaponline.com/hr/article-pdf/28/1/1/3737/1.pdf`.

Lundberg, A., Koivusalo, H., 2003. Estimating winter evaporation in boreal forests with operational snow course data. Hydrological Processes 17, 1479–1493. doi:`10.1002/hyp.1179`.

Malek, S.A., Avanzi, F., Brun-Laguna, K., Maurer, T., Oroza, C.A., Hartsough, P.C., Watteyne, T., Glaser, S.D., 2017. Real-Time Alpine Measurement System Using Wireless Sensor Networks. Sensors 17. doi:`10.3390/s17112583`.

Napoli, A., Crespi, A., Ragone, F., Maugeri, M., Pasquero, C., 2019. Variability of orographic enhancement of precipitation in the alpine region. Scientific reports 9, 1–8. doi:`10.1038/s41598-019-49974-5`.

Rice, R., Bales, R.C., 2010. Embedded-sensor network design for snow cover measurements around snow pillow and snow course sites in the Sierra Nevada of California. Water Resources Research 46, W03537.